# A single-progenitor model as the unifying paradigm of epidermal and esophageal epithelial maintenance in mice

Gabriel Piedrafita [1,2], Vasiliki Kostiou[3], Agnieszka Wabik[1], Bartomeu Colom [1], David Fernandez-Antoran[1], Albert Herms[1], Kasumi Murai[1], Benjamin A. Hall [3✉] & Philip H. Jones[1,3✉]

In adult skin epidermis and the epithelium lining the esophagus cells are constantly shed from the tissue surface and replaced by cell division. Tracking genetically labelled cells in transgenic mice has given insight into cell behavior, but conflicting models appear consistent with the results. Here, we use an additional transgenic assay to follow cell division in mouse esophagus and the epidermis at multiple body sites. We find that proliferating cells divide at a similar rate, and place bounds on the distribution cell cycle times. By including these results in a common analytic approach, we show that data from eight lineage tracing experiments is consistent with tissue maintenance by a single population of proliferating cells. The outcome of a given cell division is unpredictable but, on average, the likelihood of producing proliferating and differentiating cells is equal, ensuring cellular homeostasis. These findings are key to understanding squamous epithelial homeostasis and carcinogenesis.

[1] Wellcome Sanger Institute, Hinxton CB10 1SA, UK. [2] Spanish National Cancer Research Centre (CNIO), C/Melchor Fernández Almagro 3, Madrid 29029, Spain. [3] MRC Cancer Unit, University of Cambridge, Hutchison-MRC Research Centre, Box 197, Cambridge Biomedical Campus, Cambridge CB2 0XZ, UK. ✉email: bh418@mrc-cu.cam.ac.uk; pj3@sanger.ac.uk

The squamous epithelia that cover the external surface of the body and line the mouth and esophagus consist of layers of keratinocytes. In the mouse epidermis and esophagus cell division is confined to the deepest, basal cell layer (Fig. 1a). On commitment to terminal differentiation, proliferating cells exit the cell cycle and migrate to the suprabasal cell layers, before being ultimately shed from the tissue surface. Cellular homeostasis requires that cells are generated by proliferation at the same rate at which they are shed. Further, to maintain a constant number of proliferating cells, on average each cell division must generate one daughter that will go on to divide and one that will differentiate after first exiting the cell cycle. However, the nature of the dividing cell population has been subject to controversy[1–5]. Resolving proliferating cell behavior is key for understanding not only normal tissue maintenance but also processes such as wound healing and the accumulation of somatic mutations in normal tissues during aging and carcinogenesis[6,7].

Whilst murine epidermis and esophageal epithelium share the same basic organization, there are significant differences between the tissues. The esophageal epithelium is uniform, with no appendages, while the epidermis is punctuated by hair follicles and sweat ducts, which form distinct proliferative compartments independent of the epidermis (Fig. 1a)[5,8–10]. The structure of the epidermis also varies with body site. In typical mouse epidermis, such as that on the back (dorsum), hair follicles are frequent but there are no sweat glands[10]. In contrast, in the mouse paw epidermis hair and, particularly, sweat ducts are common in the anterior, acrosyringial region around the foot pads, while the posterior, plantar epidermis is devoid of appendages[10–12]. The ear epidermis is different again; it has uniform columns of differentiating cells, not present elsewhere[13]. Finally, the mouse tail has the most unusual structure, being a scale forming epidermis like that of chicken legs and *Crocodillia* rather than typical mammalian skin[14–16]. This structural diversity has motivated a range of studies to define the properties of proliferating cells at each site.

Genetic lineage tracing in transgenic mice has emerged as a powerful technique for tracking the behavior of cells within tissues (Fig. 1b)[17]. This is performed in mice expressing two transgenic constructs (Fig. 2a,b). The first is a genetic switch, using a bacterial recombinase enzyme *Cre*, expressed either from a transgenic promoter or targeted to a specific gene[18]. A variety of *Cre* expressing mouse strains have been used for studies of

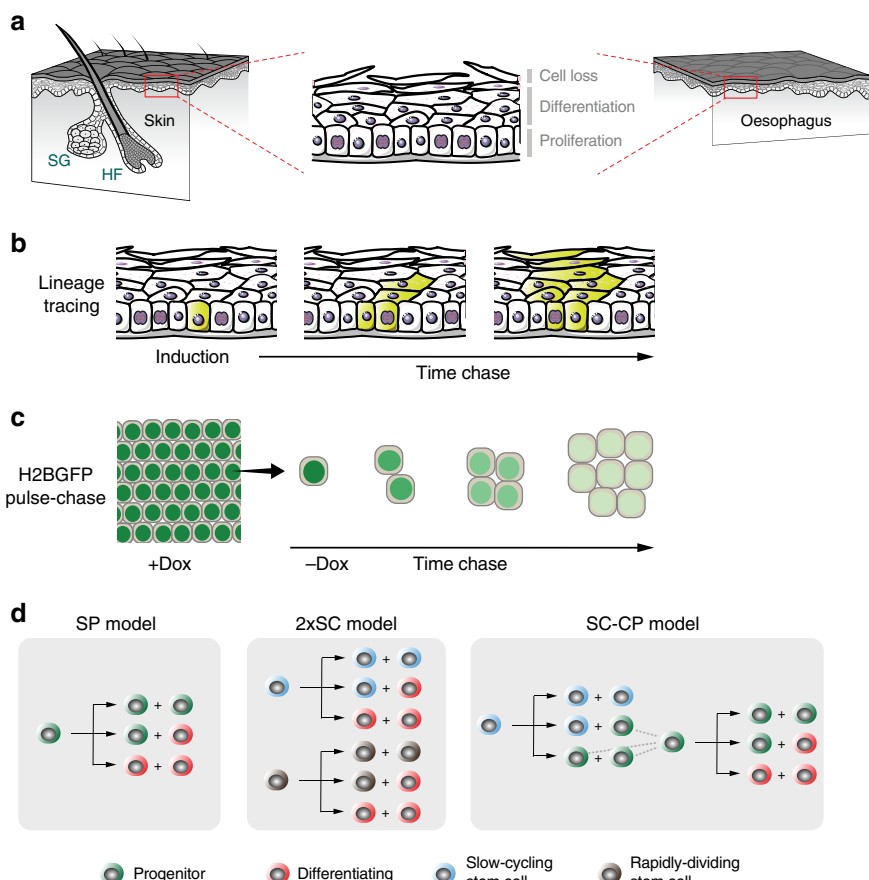

**Fig. 1 Quantitative approaches to cell behavior in murine epithelium. a** Structure of the stratified squamous epithelia from the interfollicular epidermis (skin) and esophagus of adult mice. Proliferating keratinocytes are located in the basal layer. Upon differentiation they migrate through suprabasal layers until they are ultimately lost by shedding. A balance should be established between cell division and cell loss to guarantee tissue homeostasis. HF, hair follicle; SG, sebaceous gland. **b** Rationale of genetic lineage tracing. Low-dose induction in transgenic mice allows recombination and conditional labeling of punctuated keratinocyte progenitors in the basal layer. These cells and their progeny remain labeled and can be tracked to study clonal dynamics over time. **c** Rationale of Histone 2B-GFP (H2BGFP) dilution experiments (a top-down view of the basal-layer plane is sketched). Transgenic-mouse keratinocytes express H2BGFP protein while on doxycycline (Dox) treatment. After Dox withdrawal cycling cells dilute their H2BGFP content with every division, allowing to study cell-proliferation rate. **d** Different stochastic cell-proliferation models invoked to explain epithelial self-renewal. Branches reflect different possible fates for a given proliferating cell upon division. SP: single-progenitor model. 2xSC: two stem-cell model, involving two independent types of proliferating cells dividing at different rates. SC-CP: stem cell-committed progenitor model, involving slow-cycling stem cells underpinning a second population of quickly-dividing progenitors.

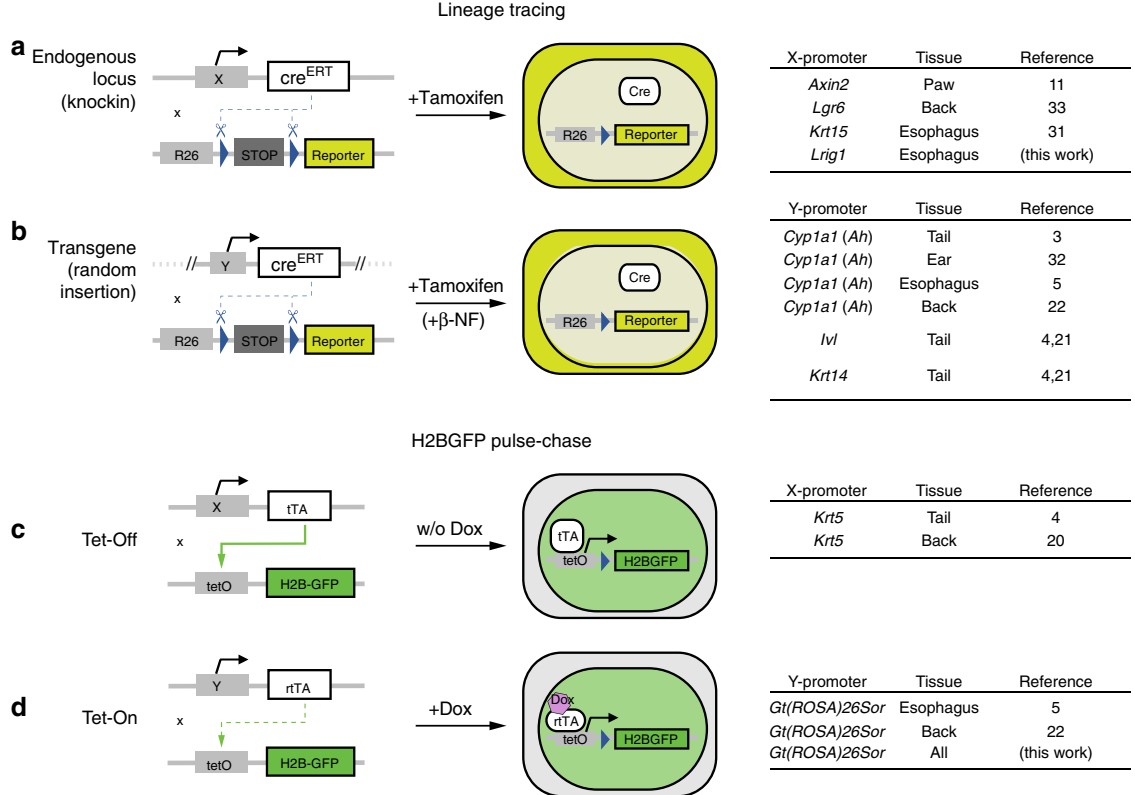

**Fig. 2 Transgenic-mouse models used for lineage tracing and cell-proliferation studies. a, b** Transgenic mice for lineage tracing are designed with two genetic constructs. The first codes for a bacterial *Cre* recombinase- mutant estrogen receptor fusion protein (CreERT), which can be targeted to a specific endogenous locus (**a**) or be under control of a transgenic promoter, randomly inserted in the genome (**b**). The second construct codes for a conditional fluorescent protein reporter, typically targeted to the ubiquitously expressed *Rosa26* locus. Treatment with tamoxifen induces Cre protein internalization to the nucleus, allowing expression of the reporter following Cre-mediated excision of a *loxP*-flanked STOP cassette. Specific details of constructs used in previous literature for lineage tracing in squamous epithelia are listed on the right. Note expression of *Cre* from the transgenic *Cyp1a1* arylhydrocarbon receptor, *Ah*) promoter requires additional treatment with a *Ah* inducer, β-napthoflavone (β-NF). **c, d** Transgenic mice for H2BGFP-dilution experiments are designed with a first construct, typically targeted to a constitutive promoter, coding either for a tetracycline-controlled transactivator (tTA; Tet-Off system) (**c**) or a reverse tetracycline-controlled transactivator (rtTA; Tet-On system) (**d**). A second construct codes for a Histone 2B-green fluorescent protein fusion (H2BGFP) controlled by a tetracycline-response promoter element (*tetO*; sometimes referred to as *pTRE*). Treatment with tetracycline or its derivative doxycycline (Dox) preempts tTA protein from binding to *tetO* elements in Tet-Off systems, causing repression of *pTRE*-controlled H2BGFP expression, whilst it is required for binding of rtTA to *tetO* elements in Tet-On systems, hence having an opposite effect. Dox is administered for induction and withdrawn during the H2BGFP-dilution chase in Tet-On mice, while in Tet-Off animals its application gets required for the duration of the experiment.

esophageal epithelium and epidermis (Fig. 2a,b). *Cre* is fused to a mutant hormone receptor so it is only active following treatment with a drug, giving control over when recombination is induced. Using low doses of inducing drug allows the labeling of scattered single cells. The second construct is a reporter, such as a fluorescent protein, typically targeted to the ubiquitously expressed *Gt(ROSA)26Sor* (*Rosa26*) locus. The reporter is only expressed following the excision of a "stop" cassette by *Cre* and expression persists in the progeny of the labeled cell. If the cells are labeled at a low frequency, single-cell-derived clones of reporter expressing cells result. If a representative sample of proliferating cells is labeled and their progeny tracked over a time course, statistical analysis of the evolving clone-size distributions may be used to infer cell behavior[3].

Alongside lineage tracing, a complementary transgenic assay may be used to detect cells cycling at different rates and infer the average rate of cell division (Fig. 1c). This uses a transgenic, drug regulated synthetic promoter to control expression of a protein comprising Histone 2B fused to green fluorescent protein (H2B-GFP) (Fig. 2c, d). The H2B-GFP is initially expressed at high levels in keratinocytes. Its transcription is then shut off and levels of H2B-GFP protein measured by microscopy or flow cytometry.

The stable H2B-GFP protein is diluted by cell division, so if the tissue contains cell populations dividing at different rates, the more slowly dividing cells will retain higher levels of protein[19]. Measurements of the rate of loss of fluorescence have been used to estimate the rate of cell division[4,5,20].

Lineage tracing has ruled out older deterministic models of a proliferative hierarchy of asymmetrically dividing stem cells generating 'transit amplifying' cells that undergo a fixed number of divisions prior to differentiation[3]. These models predict that clone sizes will rise and then remain stable. In multiple lineage tracing experiments, however, mean clone size has been found to increase progressively with time. However, several mutually incompatible models in which proliferating cells have stochastic fate have been proposed that do appear consistent with the data in one or more experiments (Fig. 1d; Supplementary Methods).

The simplest stochastic model, the single-progenitor (SP) hypothesis, proposes that all dividing keratinocytes are functionally equivalent and generate dividing and differentiating daughters with equal probability[3,5]. An alternative stem cell-committed progenitor (SC-CP) paradigm, applied to the epidermis proposes a hierarchy of rare, slowly cycling stem cells which generate stem and progenitor daughters. The progenitors

are biased toward differentiation so continual stem cell proliferation is required[4,21]. A third model argues that two independent populations of stem cells (2xSC) dividing at different rates exist in the epidermis[20]. These models all give comparably good fits to the results from individual experiments. However, each has been proposed on the basis of distinct data sets analyzed by different inference and fitting procedures, with limited testing of alternative hypotheses.

Motivated by the disparity of the proposed models of cell dynamics we set out to determine if a single model was consistent across multiple data sets in both esophagus and different epidermal regions. We use cell-cycle properties from the H2B-GFP dilution data to fit lineage tracing results by maximum likelihood parameter inference. We find that the data are consistent with a simple SP model of homeostasis. We also show that the fates of pairs of sister cells are anti-correlated, and that the basal layer contains a substantial proportion of cells which will differentiate rather than going on to divide.

## Results

**Cell-cycle times in epidermis and esophagus.** Analysis of cell proliferation in epithelia offers a simple way to test the predictions of the disparate models of epithelial homeostasis by identifying the level of heterogeneity in the division rate of basal-layer cells. The SP model predicts a single-cell population dividing at the same average rate while the alternative hypotheses argue for discrete populations dividing at different average rates. We therefore investigated the dilution of H2B-GFP in the epidermis and esophagus of $R26^{M2rtTA}/TetO\text{-}H2BGFP$ mice (Fig. 3a). The animals were treated with doxycycline (Dox) for 4 weeks to induce H2BGFP expression. Dox was then withdrawn and H2BGFP protein levels in individual basal keratinocytes tracked by direct, in situ measurement of GFP fluorescence from confocal images of epithelial wholemounts at multiple time points. We examined esophagus and epidermis from plantar area of the hindpaw, ear, and tail (Figs. 3c, 4, and 5; Supplementary Movies 1–3). Optical sections through the deepest, basal cell layer were taken over at least 5 fields of view per tissue/animal and H2BGFP fluorescence quantified for all non-mitotic nuclei following image segmentation based on DAPI staining (Fig. 3b; Methods). Non-epithelial cells in the form of CD45[+] leukocytes, which retain high levels of H2BGFP, were excluded from the analysis, but served as internal reference for label retention[5] (e.g., Fig. 3c, insert; Figs. 4b, d, f and 5b, Supplementary Data 1). In addition, for the analysis below we included a recently published data set from dorsal epidermis performed using an identical protocol[22] (Fig. 4f; Supplementary Movie 4).

We first examined images for the presence of label-retaining cells (LRCs) (Supplementary Data 1). We found no keratinocyte LRC in the basal cell layer of the esophagus or any epidermal site other than the interscale region of the tail (Fig. 5a). Rare keratinocyte LRCs (4/1923, i.e., 0.2% of basal-layer keratinocytes) were observed in interscale epidermis, in a single animal, 18 days after DOX withdrawal (Fig. 5b). Their scarcity however suggests that they are unlikely to make a substantial contribution to tissue maintenance.

Next, we performed a quantitative analysis of the time series of the individual-cell H2BGFP intensity histograms (Supplementary Data 2). If there were multiple subpopulations of cells proliferating at different rates the distribution of H2BGFP intensities would progressively diverge, becoming wider over time. We found no evidence of such behavior in the esophagus and at multiple sites in the epidermis (Fig. 3c; Supplementary Fig. 4A, B, D, F). Specifically, several statistical tests of the modality of the distribution were applied, showing no evidence for multiple

populations (Figs. 3d, 4; Supplementary Data 3; Supplementary Methods).

To further challenge the hypothesis that there is a single proliferating cell population, we examined whether this model can recapitulate the observed H2BGFP intensity distributions at each time point. For a given average division rate, we performed simulations of H2BGFP-dilution kinetics under a wide range of possible underlying (Gamma) distributions for individual cell-cycle times (Figs. 3e, 4b, d, f, Supplementary Methods). We find that the form of the H2BGFP histograms over time can indeed be fully described by a single population of cells, dividing within a relatively narrow range of cell-cycle times, further supporting the SP model (Figs. 3c, f, g, 4; Supplementary Data 4).

Altogether, these observations strongly argue against scenarios of heterogeneous proliferating cell populations, such as the SC-CP or 2xSC models, at all sites other than in the tail where marked variation between animals precluded reliable inference on cell-proliferation rates (Fig. 5c). We conclude that in the basal layer of the epidermis at multiple body sites and in the esophagus proliferating cells divide at a unique average rate with highly homogenous cell-cycle periods, consistent with the SP model (Table 1).

**A common analytical approach to resolve cell behavior.** The ability of lineage tracing to track the behavior of cohorts of proliferating cells and their progeny over time courses extending to many rounds of cell division offers the potential to validate models of homeostasis. Having established the homogeneity in the division rate of basal-layer cells, we then set out to determine whether clonal dynamics across different lineage tracing data sets were consistent with the SP paradigm.

Multiple lineage tracing studies have been published but these used distinct approaches to infer models of cell behavior and did not apply the additional constraint imposed by measuring the cell-cycle time distribution[3,4,11]. Computational simulations showed that the SP, SC-CP and 2xSC models all predict very similar development of clonal features over time, which rendered them hardly distinguishable from lineage tracing data alone (Supplementary Methods). However, as our cell-proliferation analyses do not support the SC-CP and 2xSC paradigms we focused on testing the SP model.

By incorporating the measurement of the average division rate, we could reduce the uncertainty in the parameter estimation, a problem that has been generally overlooked in these stochastic models (Supplementary Fig. 1A-B). For example, a relatively high division rate and modest proportion of symmetric division outcomes predict a similar clone-size distribution to those with a slower turnover rate but higher level of symmetric divisions. In turn, whilst long-term model predictions on clone-size distributions remained largely unaffected by the assumptions on the cell-cycle time distribution, introducing realistic estimates for the distribution of individual cell-cycle lengths affected short-term clone-size predictions, impacting on the inferred parameter values (Supplementary Fig. 2). This is due to the probability of a chain of consecutive division events deviating from the average rate, for example a run of several consecutive divisions shorter than average cell-cycle times (Supplementary Fig. 2a). This results in a broadening of the clone-size distribution at early time points after labeling. At later times, where many rounds of division have occurred in each clone, these random cycle time variations regress toward the mean cycle time of the population (Supplementary Fig. 2a). This disputes most analyses that use a Markovian implementation which makes the biologically implausible assumption that cell cycle times are distributed exponentially (i.e., the likeliest time for a cell to divide is immediately after the division

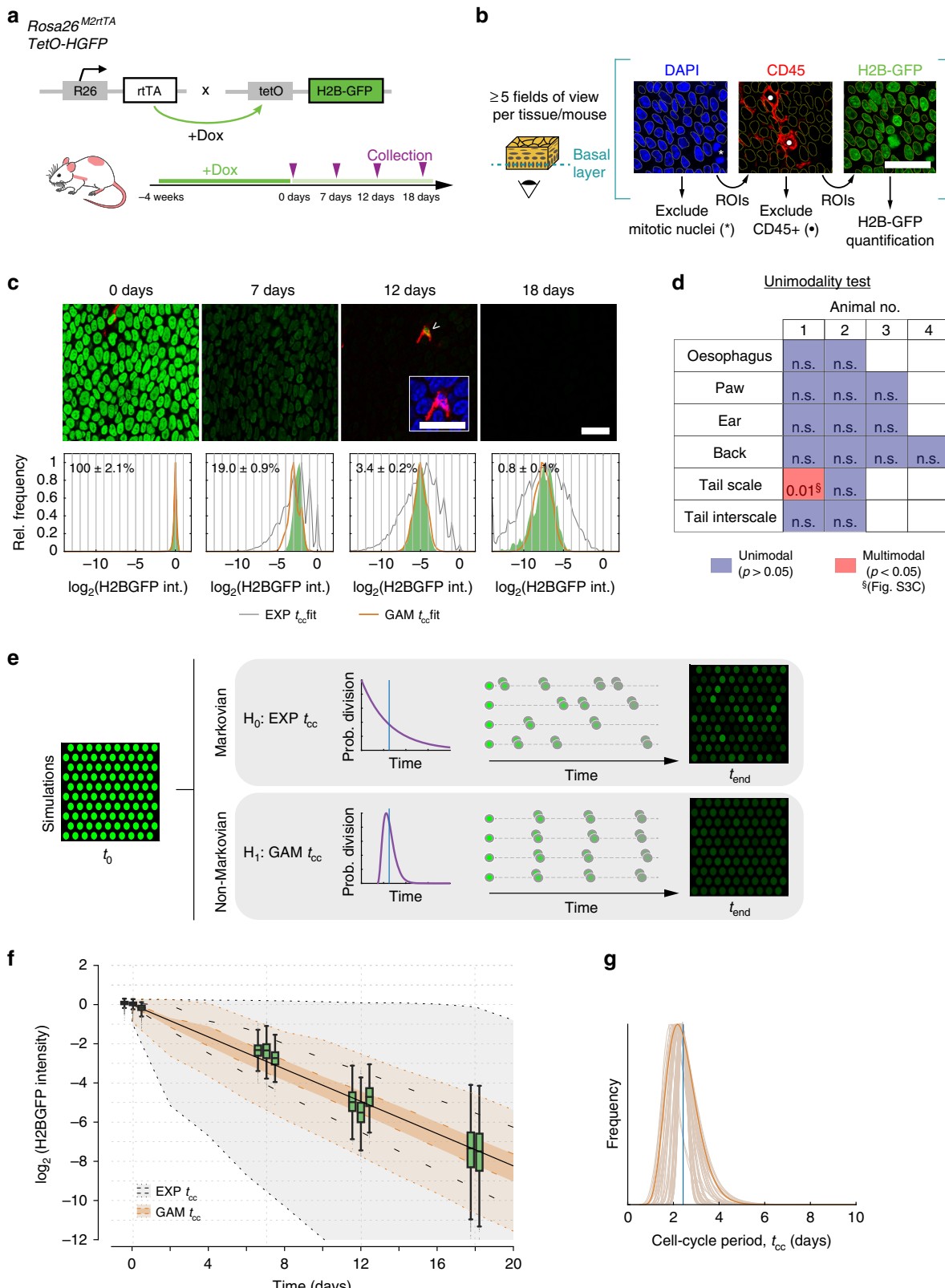

that generated it). We therefore developed a robust quantitative approach where cell-cycle attributes estimated from H2B-GFP experiments were embodied in (non-Markovian) model simulations, and a subsequent maximum likelihood estimation (MLE) method was applied across the available data sets for each body site to challenge whether each of them was consistent with the SP paradigm (Fig. 6; Supplementary Methods).

**Clonal dynamics in esophageal epithelium**. In order to explore in vivo clonal dynamics, we began by studying a lineage tracing data set from mouse esophageal epithelium (Fig. 7a). In this experiment we used a strain (*Lrig1-cre*) in which a tamoxifen-regulated form of *Cre* recombinase and enhanced green fluorescent protein (EGFP) are targeted to one allele of the *Lrig1* locus[8,23–25]. We found LRIG1 protein was ubiquitously expressed

**Fig. 3 Analysis of cell proliferation in epidermis and esophageal epithelium. a** Protocol: $R26^{M2rtTA}/TetO\text{-}H2BGFP$ mice were treated with doxycycline (Dox) to induce H2BGFP expression (green). Following Dox withdrawal, H2BGFP transcription ceases and protein levels dilute with cell division. **b** H2BGFP fluorescence was quantified in non-mitotic basal cell nuclei in optical sections of the basal layer of wholemounts. Scale bar, 20 μm. **c** Representative confocal z stacks of the esophageal basal layer showing H2BGFP (green) and pan-leukocyte marker CD45 (red). Images are representative of a total of 15 fields of view from 3 individual biologically independent mice at 0, 7, and 12 days and 10 fields of view from two individual mice at 18 days. Infrequent label-retaining cells (LRCs) (arrowhead) are positive for CD45 (insert; blue: DAPI). Scale bars, 20 μm. Histograms show keratinocyte H2BGFP intensity for each time point (in green; bottom panels) with mean ± s.e.m. values from each field of view. Best fits for the SP model with exponential- (gray) or gamma-distributed cell-cycle periods (orange lines) are shown. Raw values for H2B-GFP intensity are given in Supplementary Data 2. **d** Outcome of Silverman's unimodality test applied to individual-cell H2BGFP distributions at 18 days in esophagus and epidermis (analyses are separated per animal; this test is effectively two tailed, no multiple-testing corrections are made on p values, exact p values are given in Supplementary Data 3)[46]. A single tissue in a single animal was found to be bimodal, in this case due to variability between fields of view where cells differed by a single round of division (see Fig. 5c). **e** SP-model simulations using different distributions of cell-cycle times $t_{cc}$ (EXP: exponential; GAM: Gamma) with the same average division rate (blue vertical line), the Gamma-shaped distributions predict a more homogeneous dilution. **f** Time course in the H2BGFP intensity distributions from esophageal epithelium (normalized to average keratinocyte intensity at time 0). Green boxplots: experimental data from $n = 3$ biologically independent mice at 0, 7, and 12 days and 2 at 18 days. Centre line of box is median value, box indicates 25th and 75th centiles and whiskers indicate minimum and maximum values. The computed average division rate $\langle\lambda\rangle = 2.9/week$; solid black line. Gray region: range of H2BGFP intensities predicted from models assuming exponentially distributed cell-cycle times (interquartile range, inner dashed black lines). Light orange region: range of H2GFP intensities inferred with a gamma cell-cycle time distribution (interquartile range in dark orange, delimited by inner dashed orange lines). **g** Most-likely (gamma) shapes for the distribution of the cell-cycle period of esophageal keratinocytes, estimated from fits to the H2BGFP-dilution data. A conservative solution (in dark orange) is used for further inference. Vertical blue line: mean cell-cycle period.

in the basal layer of esophageal epithelium in wild type mice (Supplementary Fig. 3A). Consistent with this finding, in *Lrig1-cre* animals, EGFP, reporting *Lrig1* transcription was detected in 94 ± 0.3% (s.e.m.) of basal cells (Supplementary Fig. 3B). These observations indicate *Lrig1* is widely expressed in the proliferative compartment of esophageal epithelium and is suitable for lineage tracing of proliferating esophageal keratinocytes (Supplementary Methods).

To track the fate of basal cells, *Lrig1-cre* mice were crossed with the *Rosa26^{flConfetti/wt}* (*Confetti*) reporter strain which labels cells with one of four possible fluorescent proteins (green, GFP, cyan, CFP, yellow, YFP, or red RFP) after recombination (Supplementary Fig. 3c, d)[26,27]. In some *Cre* inducible mouse lines, reporter expressing clones can appear without induction with Tamoxifen. However, no fluorescent protein expression was found in adult uninduced *Lrig1-cre/Confetti* mice (Supplementary Data 5)[28]. Next, cohorts of *Lrig1-cre/Confetti* animals were treated with a low dose of Tamoxifen that resulted in labeling of only 1 in 300 ± 106 (mean ± s.e.m.) basal cells at 10 days post induction. Clones containing one or more basal cells were imaged in esophageal epithelial wholemounts from at least three mice at multiple time points over 6 months following induction (Fig. 7b; Supplementary Fig. 4a; Supplementary Data 5). Only CFP, YFP, and RFP expressing clones were counted because of *Lrig1*-driven GFP expression in all basal cells.

The pooled *Confetti* clone data set displayed several important features, which were recapitulated by clones labeled with each individual reporter. No statistically significant differences were observed between CFP, YFP, and RFP clone-size distributions at each time point (see Methods). The density (clones/area) of labeled clones decreased progressively, consistent with clone loss through differentiation, while the number of basal and suprabasal cells in the remaining clones rose (Fig. 7c; Supplementary Fig. 4B, C). The proportion of labeled basal cells remained constant during the experiment, indicating the labeled population was self-maintaining over a 6-month period, consistent with labeled cells being a representative sample of all proliferating cells in the homeostatic tissue (Fig. 7c). At late time points, the clone-size distribution scaled with time. This means that if, for example, time doubles, not only the average clone-size shape and breadth of the clone-size distribution also double. More formally, the probability of seeing clones larger than $x$ times the average clone size became time-invariant, following a simple exponential $f(x) = e^{-x}$ (Supplementary Fig. 4D)[29]. Collectively, these features are

hallmarks of neutral competition, in which clonal dynamics result from stochastic cell fates, with an average cell division generating one proliferating and one differentiating daughter cell, a scenario consistent with the SP model (Supplementary Methods)[3,29,30].

The measurement of the average cell division rate (λ) and inference of the cell-cycle time distribution constrain the fitting of lineage tracing data, providing a stringent test of the candidate SP model (Supplementary Figs. 1A–C, 2E). Within this paradigm unknown parameters are the probability of a progenitor cell division generating two dividing (PP), or two differentiating (DD) daughters (r), and the stratification rate (Γ), which in homeostasis sets the fraction of progenitor cells in the basal layer (ρ) (Fig. 6). Our technique for identifying the most appropriate cell-cycle distribution coupled with an MLE grid search estimated parameter values that gave an excellent fit with the clone-size distributions at both early and late time points for the *Lrig1/confetti* data set (Fig. 7d–e; Table 1). The model predictions were within the 95% confidence interval of the measured proportion of clones of a given size at each time point in 27/28 cases. To quantify the quality of the fit, we calculated both the determination coefficient between the model prediction and measured clone sizes, averaged across all time points, $R_T^2$, and the standard error of the fit, $S_T$, a measure of the standard deviation between the model estimates and the experimental data, averaged over all time points. For the fit of the SP model to the *Lrig1/Confetti* data set, $R_T^2 = 0.93$, $S_T = 4.3$. Values of $R^2$ and $S$ for experimental data at each time point are given in Supplementary Data 4.

Next, we applied the same approach to an independent, published lineage tracing data set from esophageal epithelium where clones were labeled with YFP by *Cre* expressed from an inducible *Cyp1a1* (*Ah*) promoter in *AhYFP* mice[5]. Parameter values very similar to those from the *Lrig1/confetti* experiment gave predictions from the SP model within the 95% CI for all 49 points in the experimental data set (Fig. 7d, e; Supplementary Fig. 4e; Table 1; Supplementary Data 4; Supplementary Methods). Quantifying the quality of fit, we found $R_T^2 = 0.98$, $S_T = 2.8$. We noted that including the cell-cycle time constraints resulted in an improved agreement with early time point clone sizes compared with the original publication ($R_T^2 = 0.97$, $S_T = 3.3$), where cell-cycle time distributions were assumed exponential (see Supplementary Data 4 for detailed goodness-of-fit statistics)[5].

As a further validation, we tested the predictions of the SP model against a third, more limited data set from *Krt15-cre^{PR1}* $R26^{mT/mG}$ mice in which a red-to-green fluorescent reporter was used with

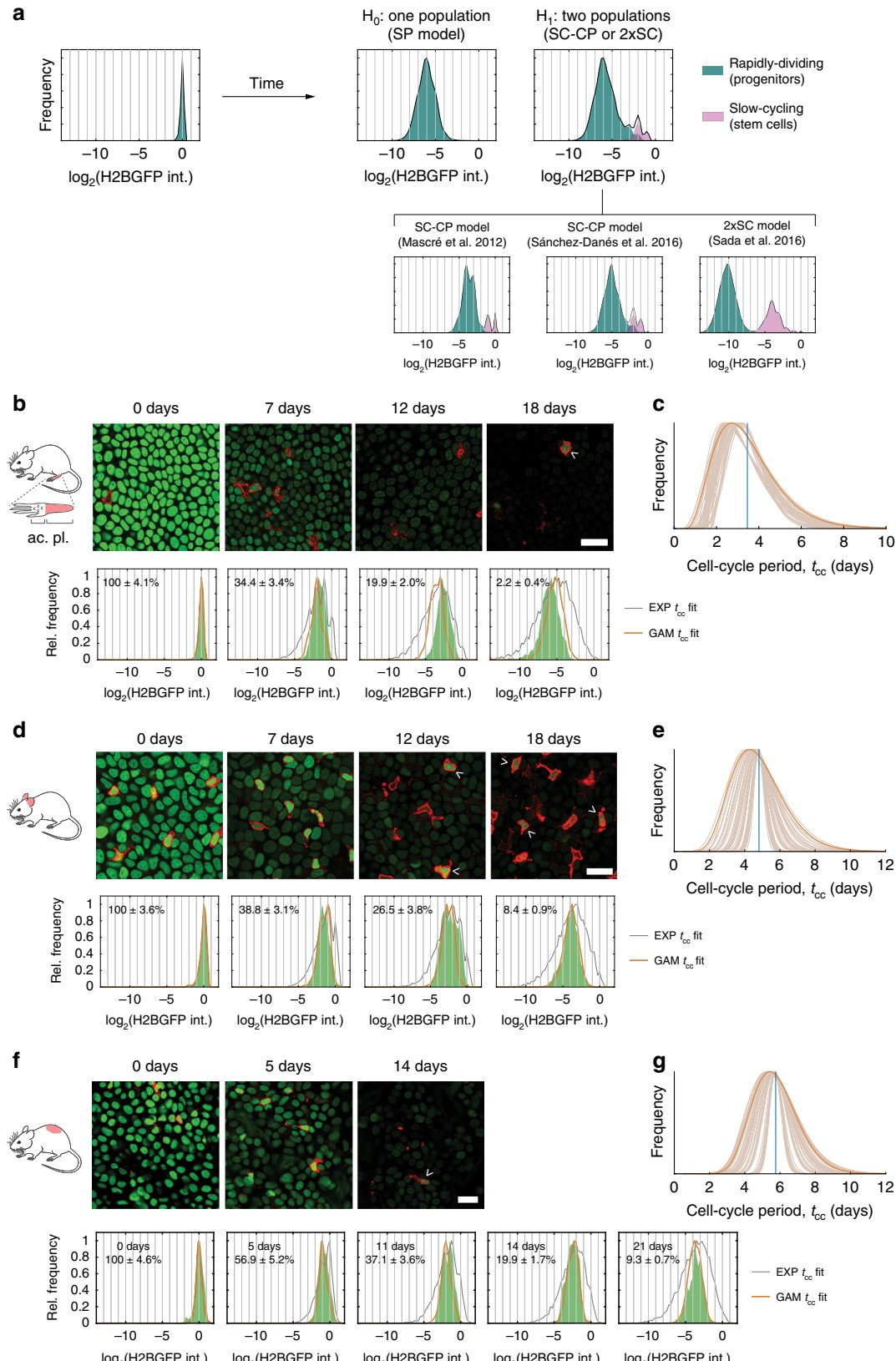

inducible *Cre* expressed from a *Krt15* promoter (Supplementary Methods)[31]. Although the SP paradigm was criticized by these authors, it yielded an adequate fit ($R_T^2 = 0.91$, $S_T = 2.6$) with their own data over the experimental time course (Supplementary Fig. 4f; Supplementary Data 4). The consistent agreement of the SP model to three independent lineage tracing data sets using different combinations of transgenic *Cre* and reporter alleles strongly supports the conclusion that the esophageal epithelium is maintained by a single-progenitor population and argues for the reliability of our parameter estimates.

**Fig. 4 H2BGFP-dilution and cell-cycle inference in skin epidermis at different sites. a** Theoretical distributions of individual-cell H2BGFP intensities expected after 3 weeks dilution under the SP, SC-CP and 2xSC scenarios assuming gamma-distributed cell-cycle periods. Simulations considered an average division rate for stem cells 4× slower than for progenitors in SC-CP and 2xSC (top panels). Predictions of each model using published parameters are shown below. All theoretical SC-CP or 2xSC scenarios represented in this figure were significant by 6 different unimodality tests (see Supplementary Data 3). **b, d, f** Representative confocal z stacks of hindpaw (plantar), ear and dorsal epidermal basal layer, respectively, from $R26^{M2rtTA}/TetO-H2BGFP$ mice, showing H2BGFP (green) and immunostaining for pan-leukocyte marker CD45 (red). Analysis of hindpaw epidermis was confined to the posterior, plantar region (pl.), excluding the acrosyringia (ac.), cartoon. Label-retaining cells (LRCs) are CD45+ leukocytes (arrowheads). Scale bars, 20 μm. Images shown are representative of a total of 18 fields of view from 3 mice at 0 and 12 days, 14 fields of view from 3 mice at 7 and 18 days in (**b**), 18 fields of view from 3 mice at 0, 7 and 12 days and 12 fields of view from 2 mice at 18 days in (**d**), and 20 fields of view from 4 mice at 0 and 11 days and 17 fields of view from 4 mice in (**f**). Lower panels: Individual-keratinocyte H2BGFP intensity levels (in green) with mean ± s.e.m. values from different fields of view. Best fits for the SP model with exponential- (gray) or gamma-distributed cell-cycle periods (orange lines) are shown. See Supplementary Data 2 for raw intensity values and summary statistics. **c, e, g** Best estimates for the (gamma) distribution of the keratinocyte cell-cycle times in hindpaw, ear, and dorsal epidermis, respectively, estimated from fits to H2BGFP-dilution data. Conservative solutions (in dark orange) are used for further inference. Vertical blue lines: average cell-cycle period per site: $\langle \lambda \rangle$ = 2.0, 1.5, 1.2/week for paw (**c**), ear (**e**), and dorsum (**g**), respectively.

**Clonal dynamics in skin epidermis**. We next investigated clonal dynamics in the epidermis through available lineage tracing data sets from the typical interfollicular epidermis of the mouse hindpaw (plantar), ear and back (dorsum). Applying the MLE approach constrained by the cell-cycle time analysis at each body site yielded slightly improved fits of the SP model to data from $Axin2$-$cre^{ERT}R26^{Rainbow}$ animals in paw epidermis ($R_T^2 = 0.98$, $S_T = 2.8$) and also ear epidermis ($R_T^2 = 0.97$, $S_T = 3.5$) and dorsal interfollicular epidermis ($R_T^2 = 0.94$, $S_T = 5.0$) from $AhYFP$ mice compared with those fits reported in the original publications ($R_T^2 = 0.93$, $S_T = 5.16$; $R_T^2 = 0.97$, $S_T = 3.52$; $R_T^2 = 0.92$, $S_T = 6.01$; respectively) (Fig. 8; Supplementary Methods, Supplementary Data 4)[11,22,32]. Despite differences in average keratinocyte division rates across territories ($\lambda \approx 2.0$, 1.5, 1.2/week for plantar hindpaw, ear and dorsum, respectively), all analyzed regions share comparable intermediate proportions of progenitor basal cells $\rho$ (~55%, the rest corresponding to differentiating basal cells) and a predominance of asymmetric cell divisions (i.e., low inferred values for the probability of symmetric division, $r < 0.25$) (Table 1; Supplementary Data 4).

Particularly relevant are the implications for the mode of keratinocyte renewal in back skin, as a previous work claims that two stem cell populations dividing at different rates coexist at this site (2xSC model)[20]. This argument was supported by a quantitative analysis of H2BGFP-dilution patterns in $Krt5^{tTA}/pTRE$-$H2BGFP$ mice, a system that differs from that we use above in that mice are treated with Dox to suppress H2B-GFP expression instead of using it as activator (Supplementary Fig. 5A)[20]. However, in that publication, which rejected the SP model, exponential distributions for cell division/cell stratification rates were assumed. Here we have shown this to be inappropriate for the short time scale of the experiment (Supplementary Methods). Our computational reanalysis, constrained by cell-cycle time distributions demonstrated the SP model gave as good a fit to the $Krt5^{tTA}/pTRE$-H2BGFP-dilution data as the more complex 2xSC hypothesis (SP $R_T^2 = 0.85$, $S_T = 0.06$ vs. 2xSC $R_T^2 = 0.87$, $S_T = 0.05$ for basal layer; SP $R_T^2 = 0.78$, $S_T = 0.07$ vs. 2xSC $R_T^2 = 0.79$, $S_T = 0.06$ for spinous layer) (Supplementary Fig. 5B; Supplementary Methods; Supplementary Data 4). Indeed, the inferred parameter values from the $AhYFP$ mouse back skin epidermis proved robust, providing good fits to another lineage tracing data set from the same body site in $Lgr6$-$eGFPcre^{ERT}Rosa26^{flConfetti}$ mice ($R_T^2 = 0.96$, $S_T = 2.39$) (Supplementary Fig. 5C; Supplementary Methods; Supplementary Data 4)[33].

Finally, we turned to revisit clonal dynamics in the mouse tail epidermis. Previous studies of tail have argued that the hierarchical SC-CP paradigm applies to proliferating cells in the interscale areas while the SP paradigm describes behavior in the scale regions (Fig. 5a)[4,21]. These claims were primarily supported by the observation of LRCs in the interscale region in H2BGFP-dilution experiments in $Krt5^{tTA}/pTRE$-$H2BGFP$ mice. However, our quantitative reanalysis of this data set showed the SP-model fits the reported H2BGFP intensity histograms over time as well as the SC-CP model (SP $R_T^2 = 0.89$, $S_T = 0.04$ vs. SC-CP $R_T^2 = 0.89$, $S_T = 0.04$) (Supplementary Fig. 6A; Supplementary Data 4)[21]. Even though we cannot discard the possibility of a subpopulation of slow-cycling stem cells in the tail, such cells would seem to be rare in interscale epidermis (Supplementary Data 1). We noted that a large proportion of the rare LRCs were identified as CD45 expressing leukocytes in our data set (Fig. 5b; Supplementary Data 1). Further analysis argued that there was no conflict between the reported tail lineage tracing data and the SP model (Supplementary Fig. 6B-H; Supplementary Methods; Supplementary Data 4).

## Discussion

Overall, we find that combining cell-cycle distribution analysis with lineage tracing argues mouse esophageal epithelium and epidermis are maintained by a single population of progenitor cells, with the sole possible exception of the interscale compartment of tail skin. The quality of the fit of the SP model to the data is equivalent to or exceeds that of more complex models, rendering the need to invoke additional cell populations redundant. The nine lineage tracing data sets analyzed include a variety of $Cre$ and reporter strain combinations, and are all consistent with the SP model. In addition, live imaging studies of the epidermis are consistent with a single proliferating cell population maintaining the tissue[34].

Quantitative analysis of cell proliferation in the different tissue types identifies further constraints that must be considered by researchers exploring the appropriateness of alternative models. The original SC-CP and 2xSC models invoked 12% and 30% of basal-layer keratinocytes constitute slow-cycling stem cells, respectively[4,20]. Histone dilution experiments have allowed us to make strong statements about the nature of any proposed second population. For each body site, with the exception of tail interscale epidermis, no keratinocyte label-retaining cells were detected in over 2000 cells imaged at each location. It follows that any slow-cycling stem-cell population must have substantially fewer than one slow-cycling cell per thousand basal keratinocytes to be compatible with observations reported here, making it unlikely that such slow-cycling cells will make a detectable contribution to tissue homeostasis. The hypothesis that two subpopulations exist, but that they both divide at a similar rate, is hard to sustain in the face of the close agreement of the simpler SP model across all the analyzed data sets.

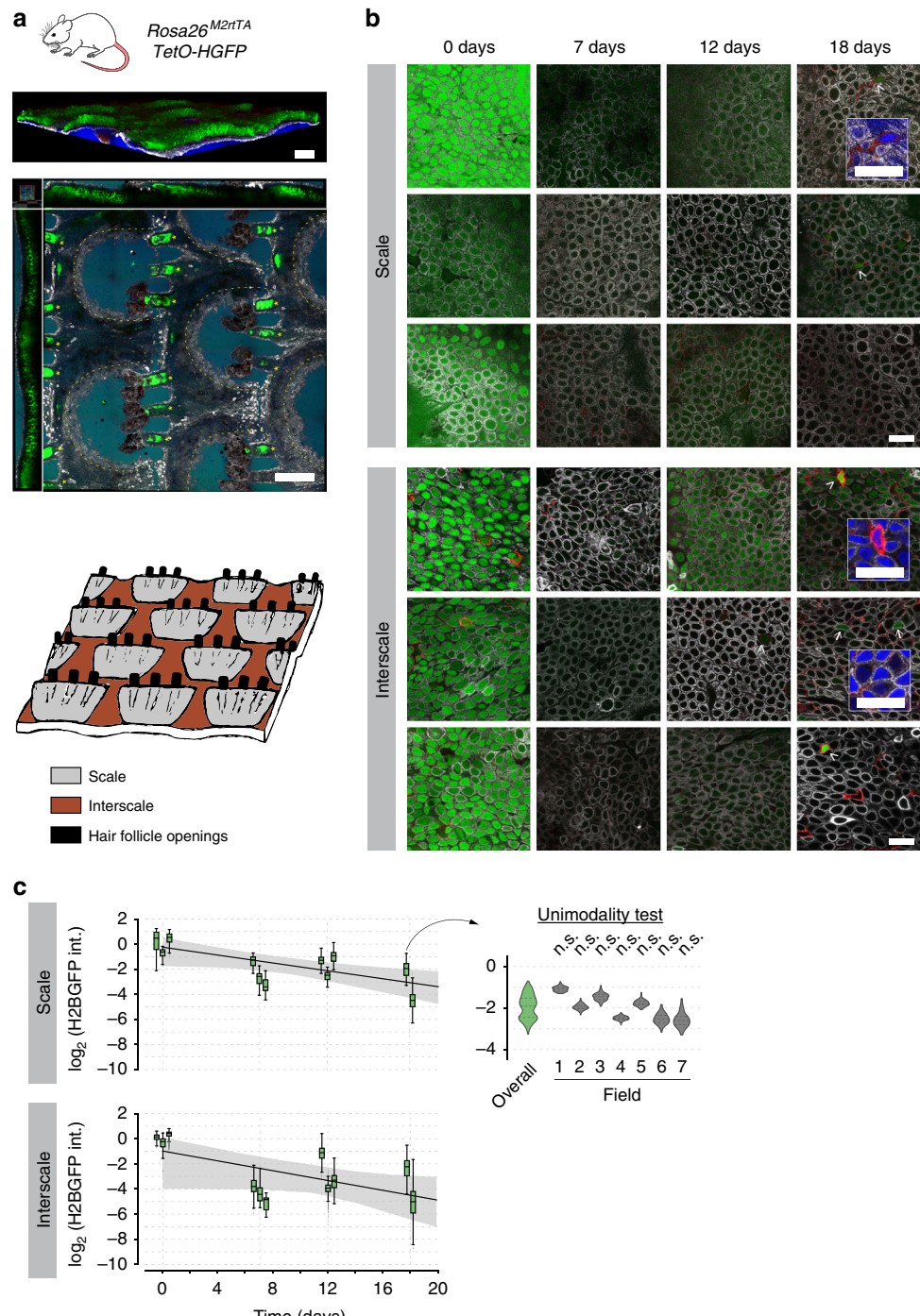

**Fig. 5 H2BGFP-dilution analysis in tail epidermis. a** Structure of the mouse tail epidermis. 3D reconstruction of confocal *z* stacks (top panel) and orthogonal xyz view (mid panel). Apical blister-shaped regions (scale) alternate with deeper regions (interscale) (boundaries delineated by dotted lines), being arranged in lines separated by hair follicles (asterisks). Green: H2BGFP expression; white: KRT14 immunostaining (as a marker of basal layer); red: CD45 immunostaining; blue: DAPI. Scale bars, 200 μm. Bottom panel: Cartoon illustrating the tail skin structure. **b** Representative confocal z stacks of scale and interscale regions of tail epidermis during H2BGFP chase experiments in *R26^{M2rtTA}/TetO-H2BGFP* mice, showing H2BGFP (green), immunostaining for KRT14 (white) and immunostaining for pan-leukocyte marker CD45 (red). Images from the same time point correspond to different mice to illustrate inter-animal variation. Label-retaining cells (LRCs) are highlighted with arrowheads (CD45+ cells) or full arrows (CD45− cells) (details in inserts; blue: DAPI). Scale bars, 20 μm. **c** Time course of basal-layer keratinocyte H2BGFP intensity distributions from scale and interscale regions of tail. Experimental data shown as boxplots per individual biologically independent mouse (intensities normalized to average keratinocyte intensity at time 0, raw H2B-GFP intensity values are given in Supplementary Data 2). *n* = 3 animals at each time point except 18 days where *n* = 2 mice. Centre line of box is median value, box indicates 25th and 75th centiles and whiskers indicate minimum and maximum values. Solid black lines: average H2BGFP-dilution rates (within 95% CI limits—shaded gray areas—where the value of *λ* = 1.2/week reported by Mascre et al.[4] falls). Insert: Detail of H2BGFP intensity distributions separated per field of view for the single tissue found to be bimodal in the overall, per-animal modality test (see Fig. 3d). Analyses per field of view all resulted non-significant (unimodality).

**Table 1 Parameter values inferred for progenitor cell behavior in different murine epithelial regions as derived from quantitative lineage tracing.**

| Tissue | Experimental model | Reference | min tcc (days) | Division rate, $\lambda$ (/week) | Symmetric division prob., $r$ | % of progenitor cells, $\rho$ | Stratification rate, $\Gamma$ (/week) |
|---|---|---|---|---|---|---|---|
| Esophagus | $Lrig1\text{-}eGFP\text{-}Cre^{ERT}/R26\text{-}^{flConfetti}$ | | 0.5 | 2.9 (2.7; 3.0) | 0.10 (0.07; 0.15) | 65 (50; 96) | 5.4 (2.9; 69.6) |
| | $Ah\text{-}Cre^{ERT}/R26^{flEYFP}$ | 5 | 0.5 | 2.9 (2.7; 3.0) | 0.06 (0.04; 0.10) | 56 (50; 89) | 3.7 (2.9; 23.5) |
| Paw Epidermis | $Axin2\text{-}Cre^{ERT2}/R26\text{-}Rainbow$ | 11 | 1 | 2.0 (1.7; 2.3) | 0.14 (0.12; 0.17) | 53 (49; 58) | 2.3 (1.9; 2.8) |
| Ear Epidermis | $Ah\text{-}Cre^{ERT}/R26^{flEYFP}$ | 32 | 1 | 1.5 (1.2; 1.7) | 0.04 (0.03; 0.06) | 54 (47; 72) | 1.8 (1.3; 3.9) |
| Back Epidermis | $Ah\text{-}Cre^{ERT}/R26^{flEYFP}$ | 22 | 2 | 1.2 (1.1; 1.3) | 0.04 (0.03; 0.07) | 61 (55; 76) | 1.9 (1.5; 3.8) |

Parameter values indicated correspond with the maximum likelihood estimate (MLE), values in parentheses are 95% confidence bounds (see Supplementary Methods for details).

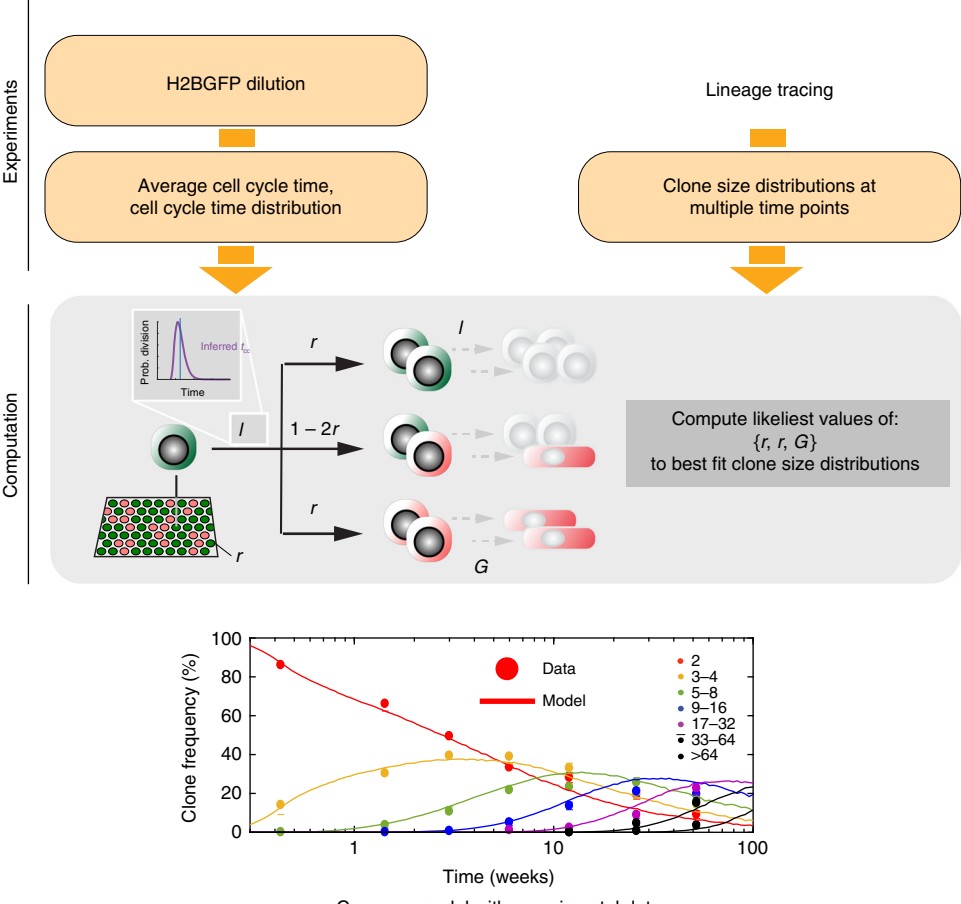

**Fig. 6 Method for single-progenitor model testing and parameter inference.** Method to single-progenitor model testing and infer model parameters. Orange boxes indicate experiments and resulting data, gray box computational model and parameter estimation. Italics indicate parameters in the SP model. The multimodality testing of H2B-GFP data showed that there is a single population dividing at the same average rate in epidermis and esophagus, consistent with the SP model (Fig. 3d). To test the SP model, the average cell-cycle time ($\lambda$) and cell-cycle time distribution were inferred from H2B-GFP experiments. These values are used in computational analysis to estimate the values of the other parameters in the SP model, the proportion of progenitor cells in the basal layer $\rho$, the proportion of symmetric cell division outcomes $r$, and the stratification rate of differentiating cells leaving the basal cell layer ($\Gamma$). Multiple sets of values for the unknown parameters were tested. For each set of unknown parameter values 100,000 progenitor-derived clones were simulated (lines) and inferred clone-size distributions compared with experimental ones (points) obtained from lineage tracing. The likeliest sets of parameter values were obtained by maximum likelihood estimation for each linage tracing data set. The quality of the fit was assessed by determining whether the simulated values lie within the 95% confidence interval of the experimental clone-size measurements at each time point.

The improved resolution of parameter estimates identifies differences in cell division rates across the epidermis and the esophagus (Table 1). Proliferating cells divide rapidly in the esophageal epithelium, on average every ~2.4 days (similar to keratinocyte turnover rate in oral mucosa), while progenitor cells cycle comparatively slower in the epidermis, on average between 3.5 and 6 days depending on body site[35]. However, our study suggests individual cell-cycle periods are tightly controlled, showing little variation around average division rate, per territory.

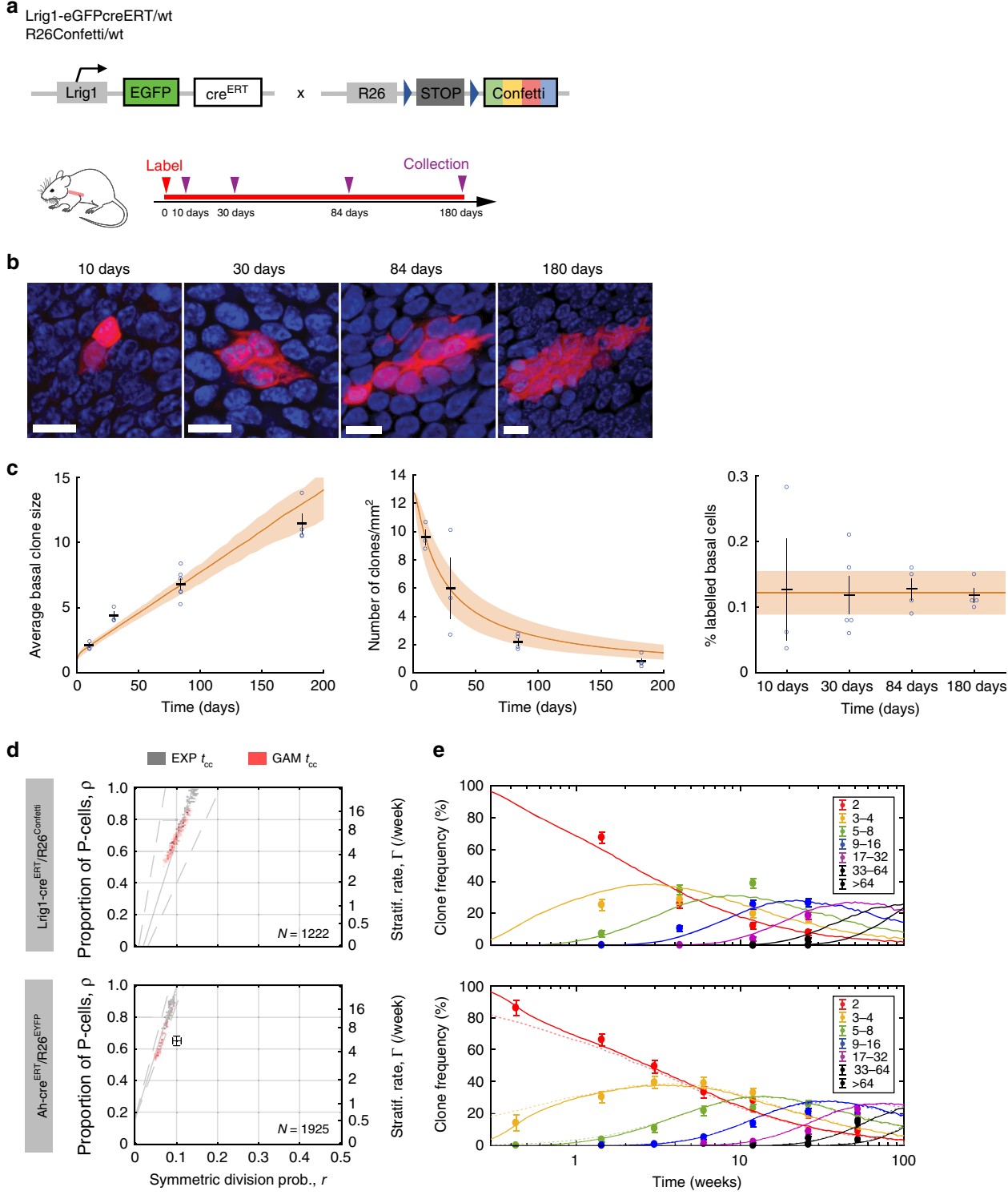

The proportion of progenitor cells in the basal layer and the probability of symmetric cell division outcomes ($r$) are similar across body sites (Table 1). The insight that a substantial proportion of cells in the basal layer will proceed to differentiate rather than divide will be important for the interpretation for the growing body of single-cell RNA sequencing data in these tissues[33,35]. In addition, the low values of $r$ we identify give insight into the basis of cell fate determination (Fig. 9). In principle, if every basal cell divides or differentiates with equal probability, as proposed by Leblond, $r$ will be 0.25, as expected from any pair of uncorrelated basal cells[1]. However, this scenario is excluded by

our analysis. Instead, the consistent values of $r < 0.25$ indicate the fate of sister cells is preferentially anti-correlated. This phenomenon can be associated to local coordination of neighboring cell stratification and division events[36]. Our results argue that anti-correlation of sister cell fates applies generally in the epidermis and esophagus, pointing to common mechanisms of keratinocyte cell fate regulation.

The single-progenitor model captures the average behavior of progenitor cells during homeostasis. However, epithelia are frequently subject to wounding. To repair the tissue requires a temporary imbalance in cell fate, with the progenitors close to the

**Fig. 7 Quantitative lineage tracing in esophageal epithelium. a** Protocol: clonal labeling was induced in *Lrig1-eGFPcre^ERT/wt R26^flConfetti/wt* mice and samples analyzed at different times from 10 to 180 days post induction, as single labeled cells develop into clones. See Supplementary Data 5 for source data for panels (**c**) and (**e**). **b** Rendered confocal *z* stacks of the esophageal basal layer showing typical RFP clones (red) at the times indicated. Blue is DAPI. Scale bars, 10 µm. Images are representative of 104 RFP clones (10 days), 75 RFP clones (30 days), 106 RFP clones (84 days), and 274 RFP clones (180 days). **c** Quantitative characteristics of the labeled clone population over time: average basal-layer clone size (i.e., mean number of basal cells/ surviving clone) (left panel), average density of labeled clones in the basal layer (middle panel), average fraction of labeled basal cells at the indicated time points (right panel). Observed values are shown in individual biologically independent mice (blue circles, *n* mice = 3 at 10 and 30 days, 6 at 84 days, and 4 at 180 days) with error bars (black) indicating mean ± s.e.m. of all mice at each time point. A total of 300 or more clones was quantified at each time point. Orange lines: SP-model fit (shaded area corresponds with 95% plausible intervals). Orange line and shading in last panel show mean and s.e.m. across all time points (from *n* = 16 mice), which is consistent with homeostatic behavior. **d** SP-model parameter inference on *Lrig1*- and *Ah-Cre^ERT* driven lineage tracing data sets from esophagus[5]. Parameter estimates are affected by the underlying modeling assumptions on the cell-cycle period, whether default exponential cell-cycle time distributions were considered (solutions in gray) or realistic gamma distributions implemented, as inferred from the cell-proliferation analysis (solutions in orange). Regions within the dashed gray lines fall consistent with the predicted $\rho/r$ ratios from the linear scaling of the average clone size. The total number of clones counted in each data set is displayed in the corresponding graph and previous parameter estimates given in ref. [5] shown as black error bars. **e** Experimental *Lrig1*- and *Ah-Cre^ERT*-derived basal-layer clone sizes from ref. [5] (dots with error bars indicating the standard error of a proportion) fit well with the SP model, with gamma-distributed cell-cycle times (lines; prediction from maximum likelihood estimation). Dim dashed lines: fits from ref. [5]. Frequencies for each clone size (basal cell number) are shown in different colors. *n* = 3 biologically independent mice at each time point except for the *Lrig1/confetti* where *n* = 6 mice at 12 weeks and 4 mice at 26 weeks.

wound producing an excess of progenitor over differentiating daughters on average. This occurs as part of a coordinated set of responses that includes cell migration and altered cell differentiation[5,37,38]. Once the epithelial defect is resolved, the progenitors revert to homeostatic balance. In esophageal epithelium and the plantar epidermis, wound repair is achieved by progenitors alone[5,11]. In the epidermis at other sites, cells migrating from other proliferative compartments, the hair follicles and sweat ducts, may also contribute to wound healing[9,10,39,40]. The ability to transiently increase the likelihood of progenitors generating proliferating progeny provides a rapid and robust response to injury. The down side of this adjustable progenitor fate is that it may be subverted by mutations acquired during tissue aging, leading to mutant clonal expansions that may undergo malignant transformation[22,41–43].

How might these findings in mice relate to homeostasis human epidermis? Human skin differs from that of mice with many more epidermal cell layers and undulates in thickness at most body sites creating folds called rete ridges and dermal papillae[44]. Nevertheless, a population of cells with balanced stochastic cell fate generating equal proportions of proliferating and differentiating cells has been identified in a live imaging study of human keratinocytes in primary culture[37]. In vivo lineage tracing in humans is not feasible. However, human epidermis has been grafted onto immune compromised mice and injected with lentiviral vectors carrying fluorescent protein reporters. When the resulting clones were imaged 6 months later they were found to vary widely in size and shape and arise from any point in the basal layer, both in rete ridges and dermal papillae[45]. These findings are consistent with the single-progenitor paradigm, but cannot provide quantitative challenge to the model available in mice.

The lineage tracing approaches considered above have been enriched by live imaging studies of mouse epidermis[34,36]. Whilst lineage tracing resolves the average behavior of a population of proliferating cells over many cell generations, live imaging allows the fate of individual cells to be resolved. Insights gained from live imaging include showing that cell fate is stochastic, the probability of generating progenitor and differentiated daughters is equal and that the fate of cells is not coordinated across cell generations, all of which are key features of the SP model[34].

We conclude that the single-progenitor model is consistent with a large body of lineage tracing and cell-cycle data collated from multiple studies and identifies the behavior of proliferating cells that underpins epidermal and esophageal epithelial homeostasis.

## Methods

**Animals**. All experiments were conducted according to the UK Home Office Project Licenses 70/7543, P14FED054 or PF4639B40. Male and female adult mice aged 3–18 months were used for in vivo experiments. Animals were housed in individually ventilated cages and fed on standard chow and maintained in SOPF health status.

Doubly transgenic, *Lrig1-eGFPcre^ERT/wt R26^flConfetti/wt* mice on a C57/Bl6N background were generated for lineage tracing studies in esophageal epithelium, by crossing *Lrig1-eGFP-ires-cre^ERT2* mice[8] onto a *Rosa26^flConfetti* multicolor reporter line[26]. Transcription of the *Cre* recombinase-mutant estrogen receptor fusion protein (Cre^ERT) is under the control of an endogenous allele of *Lrig1*. Following induction with tamoxifen, Cre^ERT protein internalizes into the nuclei and excises a *LoxP*-flanked "STOP" cassette resulting in the expression of one of the four Confetti fluorescent reporters (YFP, RFP, CFP, or GFP). *R26^M2rtTA/TetO-H2BGFP* mice, doubly transgenic for a reverse tetracycline-controlled transactivator (rtTA-M2) targeted to the *Rosa26* locus and a HIST1H2BJ/EGFP fusion protein (H2BGFP) expressed from a tetracycline promoter element, were used for label-retaining experiments[5,22]. H2BGFP expression is induced by treatment with doxycycline (Dox) and dilution of H2BGFP protein content can be chased upon Dox withdrawal. All animals were induced at 8–12 weeks age. Cohorts of at least two or three animals per time point were culled and esophagus and/or skin epidermis collected for analysis.

**Wholemount preparation and immunostaining**. Esophageal epithelium wholemounts for lineage tracing were prepared as follows: The esophagus was cut longitudinally and the middle two-thirds of the tract was incubated for 3 h in 5 mM EDTA in PBS at 37 °C. The epithelium was then peeled away from the underlying submucosa, stretched and fixed for 30 min in 4% paraformaldehyde in PBS. Samples were stored in PBS at 4 °C until subsequent analysis. Skin pieces of ~0.5 cm² were cut and incubated for 1 h in 5 mM EDTA in PBS at 37 °C. Skin epidermis was then peeled away using fine forceps and processed as described above for the esophageal epithelium.

For staining, wholemount samples were incubated in Permeabilization Buffer (PB) (0.5% BSA, 0.25% Fish Skin Gelatin (FSG), 0.5% Triton X-100/PBS) for 15 min at room temperature (RT), then blocked in 10% goat or donkey serum/PB (according to the secondary antibody used) for 1 h at RT and incubated overnight with primary antibody at 4 °C. Primary antibodies used were Lrig1 antibody (R&D Systems, Cat. AF3688), ITGA6 antibody (clone GoH3, Biolegend, Cat. B204094), Alexa Fluor® 647 anti-CD45 (clone 30-F11, Biolegend, Cat. 103124), Keratin 14 antibody (clone Poly19053, Biolegend, Cat. 905301). Samples were subsequently washed four times for 30 min in 0.2% Tween-20/PBS and incubated with an appropriate secondary antibody for 3 h at RT. Secondary antibodies used were Goat or Donkey Alexa Fluor 488/546/555/647 (Molecular Probes). A washing step with 0.2% Tween-20/PBS was repeated and samples were incubated for 30 min with DAPI (Sigma-Aldrich) and finally mounted in DAKO Vectashield Mounting Medium with DAPI (Vector Labs).

**Dilution of Histone 2B-GFP protein content**. *R26^M2rtTA/TetO-H2BGFP* animals were treated with doxycycline (Dox, 2 mg/ml in drinking water sweetened with 10% blackcurrant & apple) for 4 weeks. Dox was then withdrawn and animals culled at different time points to track H2BGFP florescence dilution. Epithelial wholemounts from esophageal epithelium and skin epidermis were imaged on a Leica TCS SP8 confocal microscope using ×20 or ×40 objectives at 1024 × 1024 resolution, line average 4 and 400 Hz scan speed. Individual-cell H2BGFP

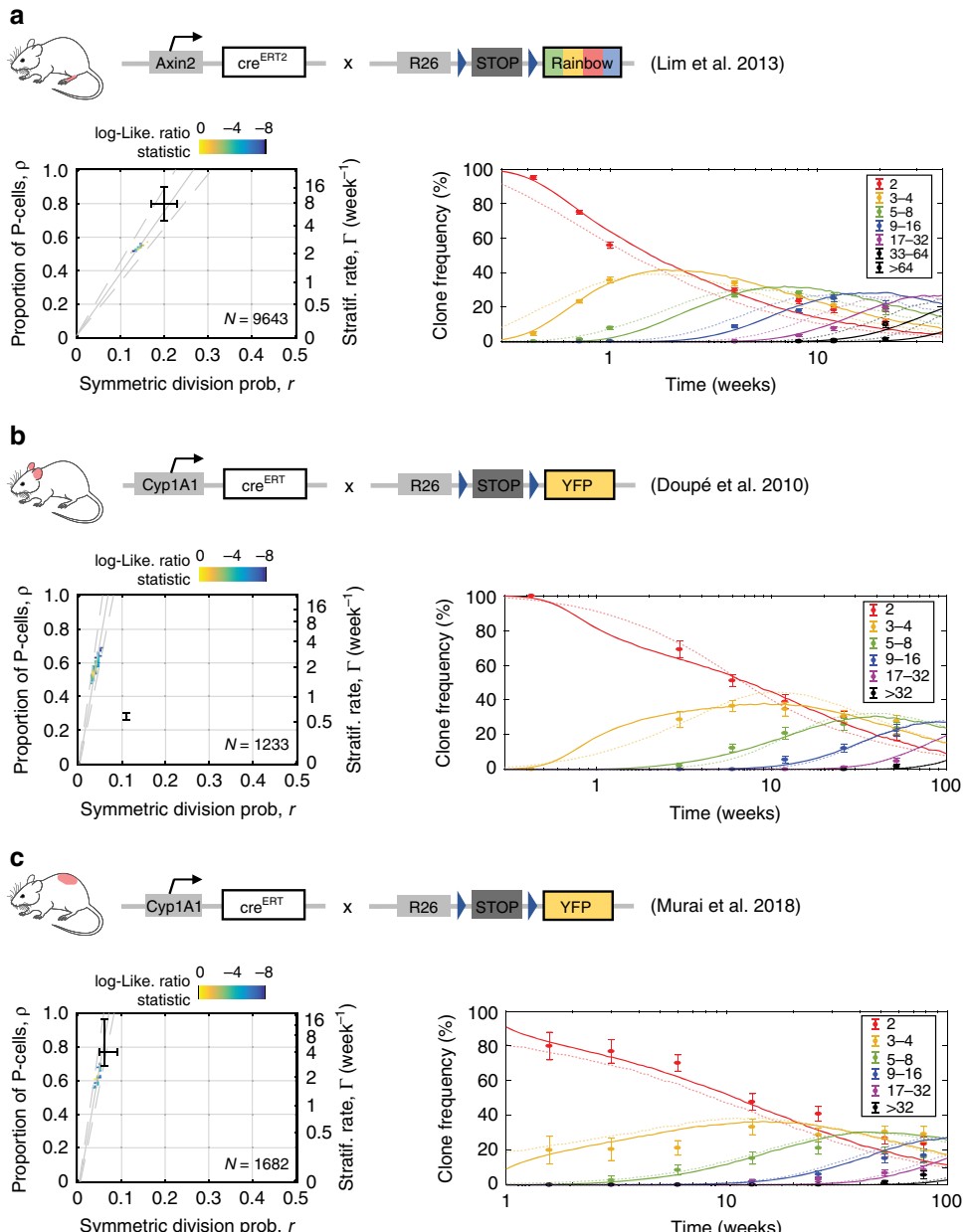

**Fig. 8 The single-progenitor model fits clone dynamics in different regions of skin epidermis. a–c** Left panels: SP-model parameter inference on lineage tracing data sets from paw epidermis[11] using *Axin2-creERTR26Rainbow* animals (**a**), and ear[32] and dorsal[22] interfollicular epidermis in *AhYFP* mice (**b** and **c**, respectively). Parameter estimates are obtained by MLE based on SP-model simulations constrained by the cell-cycle period distribution inferred from each corresponding cell-proliferation analysis. Regions within the dashed gray lines fall consistent with the predicted $\rho/r$ ratios from the linear scaling of the average clone size. The total number of clones counted in each data set is displayed in the corresponding graph. Black bars are parameter estimates given in the original publications shown, centre is the mid range and bars indicate the maximum and minimum plausible parameter values in simulations in each paper (**a**, **b**, **c**). Right panels: Experimental *Axin2-* (**a**; from ref. [11]) and *Ah-* (**b** and **c**; from refs. [32] and [22]) derived basal-layer clone sizes (dots indicate mean ± standard error of proportion) give an excellent fit with the SP model with gamma-distributed cell-cycle times (lines; prediction from MLE). Dim dashed lines: fits obtained with parameter estimates given in the original publications. Frequencies for each clone size (basal cell number) are shown in different colors. See Supplementary Data 4 for goodness-of-fit statistics.

intensities were determined by image segmentation/nuclear identification, using a semi-automated object-recognition macro (based on the DAPI channel) built in ImageJ, and the process completed by manual curation. Per-cell intensity values given are averaged over all nuclear pixels. All H2BGFP samples were stained for CD45 and positive cells excluded from the analysis.

**Lineage tracing**. Low-frequency expression of the Confetti reporters in the *Lrig1-eGFP-ires-creERT2 R26flConfetti/wt* mouse esophagus was achieved by inducing 10-week-old animals with intraperitoneal injection of a single dose of 1 mg tamoxifen (100 µl of 10 mg/ml) on two consecutive days[8]. This resulted in a labeling efficiency

of 1 in 301 ± 106 (mean ± s.e.m.) basal cells by 10 days post induction (allowing individual clone tracking without merging). Between three and six mice were culled per time point. Confocal images of immunostained wholemounts were acquired on a Leica TCS SP8 confocal microscope (×10, ×20, and ×40 objectives; typical settings for z-stacks acquisition: optimal pinhole, line average 4, bi-directional scan with 400–600 Hz speed, resolution of 1024 × 1024 pixels).

The number of nucleated basal and suprabasal cells per labeled clone was counted under live acquisition mode. GFP-labeled clones were not scored due to the difficulty of distinguishing them from the constitutive basal GFP expression driven by the *Lrig1* cassette. CFP, RFP, and YFP clones were pooled together for further analysis (histograms (distributions) of basal-layer clone sizes and average

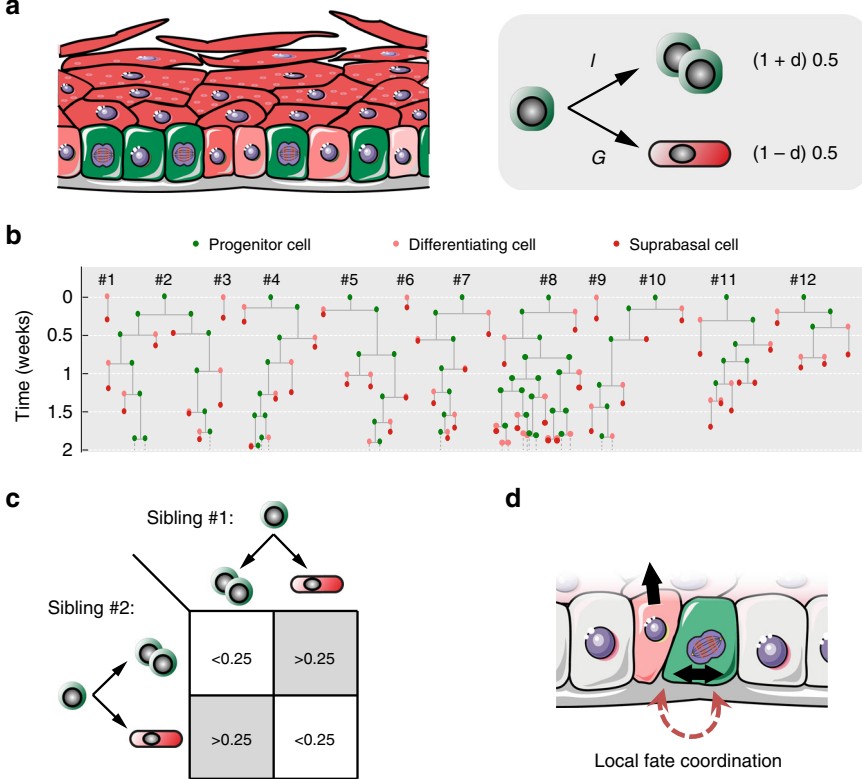

**Fig. 9 Cell fate coordination underpins single-progenitor dynamics in epidermis and esophagus. a** Epidermis and esophageal epithelium are maintained by a single population of progenitor cells. Left panel: Progenitor cells (in green) share the basal layer with post-mitotic keratinocytes in early stages of differentiation (pale red), which are transiently retained in the basal compartment before stratification. Right panel: Simplified representation of the single-progenitor model focusing on individual basal cell fates. Epithelial cell dynamics are dominated by stochastic but skewed fates through spatial coordination between neighboring or sibling cells. Individual basal cells undertake dichotomic decisions: they have the potential to divide but can alternatively differentiate exiting the basal layer. Both probabilities are balanced (50–50%) across the entire proliferating cell population, but can be decompensated or skewed for individual cells depending on their local niche, as reflected with the parameter $\delta$ (which works as a context-dependent modulatory factor). **b** Stochastic progenitor fates explain a scenario of neutral clonal competition dynamics where clones develop into heterogeneous sizes, constrained by cell-cycle time control and fate coordination effects. Displayed is a representative set of epithelial clone dynamics simulated using the parameters inferred for murine esophageal epithelium homeostasis. **c, d** Our results demonstrate that the outcome of sibling keratinocyte cells is commonly biased toward an excess of asymmetric fates where one decides to divide while the other differentiates, in agreement with a single-progenitor model with low values of $r$ ($r < 0.25$).

number of basal cells). A total of 300, 315, 302, and 305 labeled clones from 3, 3, 6, and 4 mice at 10, 30, 84, and 180 days post induction, respectively, were quantified. Regarding the time courses in the number of clones per unit area and the proportion of labeled basal cells, only RFP clones were considered given the low, variable induction of the other florescent reporters and their overall small contribution (including these numbers did not alter the conclusions).

Lineage-tracing data from *Ah-cre^ERT R26^flEYFP* derived clones in esophagus[5], ear[32], and dorsal epidermis[22] were obtained from experimental colleagues (data available upon request). Data on induced *Axin2-cre^ERT R26^Rainbow* clones in hindpaw[11] and *Lgr6-eGFPcre^ERT R26^flConfetti* in back epidermis[33] were kindly provided by the authors. Data from lineage tracing in scale and interscale tail epidermis[21] were accessed through the online publication material, while authors were unable to provide original data from ref. [4]. Data on *Krt15-cre^PR1 R26^mT/mG* mouse esophagus[31] were retrieved by digitalizing Fig. 2e and Figure S3B from the original publication. A similar procedure was used to extract *Krt5^tTA/pTRE-H2BGFP*-dilution data from back skin (Fig. 3 from ref. [20]) and tail epidermis (Fig. 3k from ref. [4]).

**Mathematical modeling and statistical inference**. Model dynamics were simulated using Markovian (Gillespie algorithm) and non-Markovian exact stochastic Monte Carlo methods implemented in Matlab. A maximum likelihood estimation (MLE) approach was followed for parameter inference, except when stated otherwise, and best-fit parameters obtained with 95% confidence intervals based on the likelihood-ratio test (alpha = 0.05). The coefficient of determination ($R^2$) and the standard error of the fit (S) were calculated for the evaluation of goodness of fit (GoF). In those cases involving data at various time points, GoF values averaged across the different time points ($R_T^2$ and $S_T$) are displayed. A comprehensive list of detailed $R^2$ and S values can be found in Supplementary Data 4. For details of theoretical modeling and computational methods used to infer cell behavior and clonal dynamics, see Supplementary Methods section below.

**Reporting summary**. Further information on research design is available in the Nature Research Reporting Summary linked to this article.

## Data availability

The authors declare that the experimental data supporting the findings of this study are available within the paper and its supplementary information files.

## Code availability

Code used in computational modeling is available in Github: https://github.com/gp10/Piedrafita_etal_SI_code/

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

## Acknowledgements

We are grateful to Xinhong Lim and Roeland Nusse; Kyogo Kawaguchi, Allon M. Klein and Valentina Greco; Anja Füllgrabe and Maria Kasper; and David Shalloway for kindly sharing raw lineage-tracing data and computational algorithms from previous publications, and for their valuable comments. We thank María P. Alcolea for l feedback on H2BGFP experiments and Kim Jensen for the Lrig1-creERT mouse strain, Esther Choolun and staff at the MRC ARES and Sanger RSF facilities for excellent technical support and Hall and Jones' group members and Fernando Pozo, Fátima Al-Shahrour, and Francisco X Real at CNIO for critical comments. This work was supported by grants from the Wellcome Trust to the Wellcome Sanger Institute, 098051 and 296194, Cancer Research UK Programme Grants to P.H.J. (C609/A17257, C609/A27326) and the Royal Society (UF130039 to B.A.H.). G.P. is supported by a Talento program fellowship from Comunidad de Madrid.

## Author contributions

G.P. performed the computational simulations and carried out the analysis of experimental results. A.W. carried out the lineage tracing experiments. B.C. and D.F.A. performed H2BGFP-dilution experiments. A.H. and K.M. contributed with image acquisition and image analysis. V.K. performed preliminary, conceptual work and helped implementing the Non-Markovian simulation algorithm. B.A.H. and P.H.J. supervised and designed the work. G.P., B.A.H., and P.H.J. wrote the paper. All authors reviewed and edited the final version.

## Competing interests

The authors declare no competing interests.
