## [Peer Review File · Nature Communications]

Reviewers' Comments:

Reviewer #1:

Remarks to the Author:

The manuscript by Peidrafita et al addresses both proliferation and clonal dynamics in a variety of mouse epithelia under homeostasis with the goal of inferring stem/progenitor cell behavior as stochastic or hierarchical. As stated by the Authors, this has become a contentious issue in the epithelial stem cell field given that different Cre drivers and physiological conditions, i.e. wounding versus homeostasis, lead to sometimes polarizing interpretations on epithelial stem cell homeostasis, particularly in the adult skin. Resolution of this issue would represent an important contribution to the field. However, as presented it is unclear whether the results presented move the needle sufficiently to resolve this issue. While the clonal analyses in all epithelia collectively support a single progenitor hypothesis, these studies are largely correlative to previous work by this group. This study does not address the larger issue of potential target cell heterogeneity exposed by different Cre drivers that may underlie the discrepancies in clonal dynamics reported by multiple labs. The data presented using the Lrig1Cre mouse supports this concern. There is also general concern, which is discussed in detail below, over the appropriateness of the H2BGFP mouse as a model for proliferation.

Specific concerns:

Figure 2

1. More explanation is required for the H2BGFP mouse model as it relates to IFE basal layer proliferation. First, the proliferative index of mouse basal IFE across a variety of genetic backgrounds is low, ranging from 2-5%. As such a maximum of 30% of basal cells would be predicted to undergo a round of proliferation by 7 days. However, Figure 2B indicates that the entire basal layer has undergone at least one proliferation event (all cells show lower GFP intensity by confocal and FACS).
2. The additional shifts in the FACS histograms at 12 and 18 days suggest more proliferation events, totaling at least three over the course of an 18-day chase. The concept of the entire basal IFE undergoing three rounds or more of proliferation over the course of 2.5 weeks seems unlikely and does not fit with established proliferative index rates. This needs to be addressed as it suggests that the transgenic model does not elaborate normal basal cell proliferation under homeostasis.
3. It is not clear whether the GFP+ population is gated prior to generation of the histogram plots shown in Fig 2B. A basal keratinocyte marker should be employed to demonstrate the percentage of basal keratinocytes that are GFP pos v. neg at each chase time point. As presented, the analysis in Fig. 2B appears to only show GFP+ cells and does not take into account whether any basal keratinocytes may be GFP negative (therefore undergone more rounds of proliferation than their GFP+ neighbors in the IFE).

Fig S6

1. The GFP intensity in the Lrig1 Cre mouse model shown in Fig S6 does appear heterogenous. As such, while endogenous Lrig1 appears uniform by tissue immunofluorescence, Cre expression may differ dramatically in basal esophageal cells. This possibility confounds the Author's interpretations over an SP model if they are preferentially targeting high-expressing Cre cells with low dose tamoxifen. FACS analysis may remedy this issue.

Reviewer #2:

Remarks to the Author:

In this study Piedrafita et al. perform statistical meta-analysis of previously published as well as newly generated experimental data to test alternative hypotheses for the cellular mechanism of homeostatic maintenance of squamous stratified epithelia. The identity of stem cell and progenitor populations and the mode by which they are replenishing stratified epithelia such as the skin epidermis and the esophagus have been subjects of intense research for the past decades. Alternative interpretations of experimental data from these studies have led to mutually exclusive hypotheses, which despite the numerous high-impact publications remain unsettled and are subject of intense debate to this date. The current study by Piedrafita et al. does not offer any new groundbreaking discoveries. However, using an integrative and rigorous quantitative approach the authors attempt to explain and reconcile the apparent disparity of proposed models. For this, they take advantage of published clonal analysis data and combine them with their own lineage tracing analyses. Furthermore, the authors perform new, elegant H2B-GFP based, label-retention experiments that provide critical constraints to further refine their mathematical model. They conclude that the vast majority of experimental data in past and present studies are consistent with a "single progenitor" model which supports the idea of an equipotent population of stem cells at the basal layer of squamous stratified epithelia that support the tissue homeostasis through stochastic but balanced fates at the population level. Despite the apparent lack of novelty, this manuscript addresses an important question in the stem cell and epithelial biology fields and provides critically needed insight and clarity that in my view (and until new evidence come to light) settles the ongoing debate. For these reasons I recommend that this manuscript is published in its current form and in a timely fashion.

- There are other squamous stratified epithelia that do not appear to adhere to the same SP model as the skin epidermis and the esophagus. One example is the ocular surface epithelium, which is also stratified but which follows an apparent hierarchical organization with long lived stem cells located at the periphery, in the limbus while transient amplified progenitors migrate into the center of the cornea. Additionally, transitional zones in the anorectal tract appear to also have unique cellular organization and activities. Other squamous stratified epithelia, like those in the reproductive tract are less studied and therefore, poorly understood. Based on these examples I feel it is prudent to avoid blanket statements about the models tested in this study and their application to all squamous epithelia.

- Fig 1. According to table 1 the estimated rate of division is 1.2/week for the back epidermis. However, at their 7 day timepoint the fluorescence intensity seems particularly low and certainly less than 50% of that in day 0? The same seems to be the case for the examples shown in Fig.S3B. Does fixation affect the intensity and subsequent quantifications? Also, how do the authors distinguish basal cells? A month of Doxy induction means that both basal and suprabasal layers are GFP+. Perhaps the authors could consider including representative serial optical sections in the form of movies accompanying the manuscript.

- The label retaining experiments in Fig2/S2 are very informative and provide critical cell proliferation parameters for the subsequent mathematical modeling. It is particularly important that the authors make a specific mention and control for the presence of GFP+/CD45+, non-epithelial cells. This apparent artifact of the H2B-GFP mouse model is often overlooked in published studies but could have significant implications for the interpretation of the data.

- Since the computational approach carries the entire novelty of the paper it should be explained more clearly. I found the figures (both supplementary, SF4 and SF5) corresponding to the section of the paper in which this is described ("A common analytical approach integrating ...") somewhat confusing. This may be primarily due to my lack of expertise in mathematical modelling but it was often unclear how the conclusions made in the text, for example that distribution of cell cycle lengths affects predictions of clonal growth in the short but not in the long term were supported by these figures. Perhaps the authors could refer to specific panels rather than the entire supplementary figure. Along the same lines, it's unclear to me why these important figures detailing the main contribution of the paper are in the supplementary section.

- Fig.S6B It's hard to tell whether that is actually a 94% positive population. Can the authors show a picture that also has DAPI staining to have a better visualization of the ubiquitous labeling of the

Lrig-CreER line?

- Fig.S6D How long after induction was this image acquired?
- When the authors reanalyzed the data from Doupe et al. (2010) they claim the original authors reached a different conclusion regarding the progenitor model the dataset supported. Can the authors explain why different conclusions were reached from the same dataset in this particular case? (similar to what is discussed later in the text for Sada et al. (2016).
- In page3 the authors state "In contrast, in the mouse paw epidermis hair is absent but sweat ducts are common". This statement is incorrect as hair follicles are present in varying densities in the mouse footpads (see Kamberov & Tabin, PNAS 2015).

Reviewer #3:

Remarks to the Author:

Piedrafita et al. provided a thoughtful analysis, and devised a revised model to explain how squamous epithelial tissues are maintained under homeostatic conditions. The authors described the cell-cycle properties of basal cells extracted by histone dilution data using an integrative stochastic model, and concluded that a single progenitor model can explain squamous epithelial tissue maintenance. Importantly, the authors demonstrate that their single progenitor model can explain previously proposed, mutually irreconcilable models of epidermal homeostasis. The theory behind the proposed model is consistent, and carefully explained. The manuscript is technically sound, and the quality of data presentation is high. These results will expand our understanding on how epithelial tissues are maintained under homeostatic conditions, and they represent a useful resource for further mechanistic studies and possible extrapolation to other tissues characterized by fast turn-over rates. Here are some minor comments:

Specific comments:

1. Can the newly revised single progenitor model be used to explain cellular competition during development and/or tissue repair? If not, what are the major new elements needed in the model? Some discussions along this line would be helpful.
2. Ghazizadeh et al. (PMID: 15675956) developed a model of how human skin epidermis is maintained using long-term lineage tracing data in xenografts. While human epidermal maintenance is not the main aspect of this paper, is the proposed model applicable to human epidermal maintenance?
3. Using live imaging approach in living mice, Rompolas et al. (PMID: 27229141) suggests that a single proliferating population maintains ear and paw epidermis. This model resonates with the model proposed herein. Does those data fit with their newly revised single progenitor model?
4. The term "good fit" was used multiple times in the manuscript. Is there any quantitative measurement for defining "good fit"?
5. In Figure 3B, no data for two clones derived from Lrig1/Confetti mice is shown for 180 days. Is there a particular reason for why these were not shown?
6. In Figure 3C, it would be useful to present: i) average clone size, ii) number of clones/mm², and iii) percentage of basal cells per individual clones (i.e. CFP, YFP, and RFP).
7. The H2BGFP dilution analysis and keratinocyte cell cycle inference in dorsal skin differs from that of paw, ear and tail (related to Supplementary Figure S2 and S3). Is there a reason why dilution experiments were performed differently in back skin compared to other skin territories?
8. Supplementary Figure S4D shows that the single progenitor model parameter discrimination improves as the sample size is increased. Here, the legend suggests that 100, 1000, and 10000 clones were sampled but figure suggests 400, 4000, and 40000. What was the sample size used?
9. In supplemental material, the "P" in Eq. (15) and Eq. (16), are they the same? The descriptions for Non-Markovian simulations are confusing.
10. What is "scaling behavior"?
11. Page 10 of main text: $f(x) = e^{-x}$ not $f(x)=e^{-x}$.

12. Page 12 of main text: Letter "F" in the middle of the page.
13. Where is the legend for subfigure E in Figure 3? It seems missing.
14. In Supplement section 2, what is cumulative frequencies? Pncum ?

Response to Reviewers' Piedrafita et al., NCOMMS-19-24313A

We are most grateful to the reviews for their careful reading of the manuscript, perceptive comments and helpful suggestions. These have significantly improved the revised version of the paper. In the following, our responses are in blue.

Reviewer #1 (Remarks to the Author):

The manuscript by Peidrafita et al addresses both proliferation and clonal dynamics in a variety of mouse epithelia under homeostasis with the goal of inferring stem/progenitor cell behavior as stochastic or hierarchical. As stated by the Authors, this has become a contentious issue in the epithelial stem cell field given that different Cre drivers and physiological conditions, i.e. wounding versus homeostasis, lead to sometimes polarizing interpretations on epithelial stem cell homeostasis, particularly in the adult skin. Resolution of this issue would represent an important contribution to the field.

We thank the reviewer for their comments and agree that this issue is of considerable importance to the field.

However, as presented it is unclear whether the results presented move the needle sufficiently to resolve this issue. While the clonal analyses in all epithelia collectively support a single progenitor hypothesis, these studies are largely correlative to previous work by this group.

Our group's work is indeed included as we have published several studies in this area. However, we have included all available quantitative datasets, several of which are in papers where the data has been interpreted as arguing against the single progenitor model. Specifically, our own group's studies cover tail, ear and back epidermis and 2 data sets from esophageal epithelium, including the *Lrig1-cre* experiment presented in this paper. The majority of data sets are from other researchers: two each on dorsal epidermis (Kasper, Tumber groups), hind paw (Nusse, Greco) and tail (Tumber, Blanpain), one on ear (Greco), and a more limited data set on esophagus (Rustgi). Our goal was to see whether all the available lineage tracing data sets including those argued to support conflicting models can be reconciled quantitatively within a single paradigm.

In addition to lineage tracing datasets, our conclusions have been drawn from a detailed statistical analysis of histone dilution data. We have shown not only that the SP model reproduces lineage tracing observations, but also that the cell division data do not support a second population of cells dividing at a different rate through the application of multi-modality analysis with several different algorithms. This is key, as the alternative models of homeostasis all invoke cell population(s) cycling at different rates). The observation that all cells share the same average division rate is key to understanding homeostasis in epidermis and esophagus.

This study does not address the larger issue of potential target cell heterogeneity exposed by different Cre drivers that may underlie the discrepancies in clonal dynamics reported by multiple labs. The data presented using the Lrig1Cre mouse supports this concern.

We agree that the difference in Cre- drivers may be a problem if they label functionally distinct target cell populations. Within a single cell type, gene expression varies widely even within a given cell type due to cell cycle and circadian fluctuations and the influence of the local microenvironment. However, such transient changes in the expression of a large number of genes need not indicate functional differences in cell behaviour. We would submit that the most reliable and sensitive assay of the functional properties of the cells labelled by each Cre line is lineage tracing of the labelled population, the evolving distribution of labelled clone sizes, over a large number of rounds of division which captures the average behaviour of the labelled cells¹⁻³. We find that despite the diversity of Cre lines in the literature, the behaviour of the cells they label is remarkably consistent with a single model of cell behaviour. Our analysis argues against the contention that there are 'discrepancies in clonal dynamics reported by multiple labs', the data from all the studies we analyse are remarkably consistent, it is the interpretation of the results that differs.

A key argument in the manuscript is that the dynamics of labelled cells are not determined by the Cre line used to label them, a finding consistent with the single progenitor hypothesis. This extends to our findings in other territories such as back skin, where clone dynamics of transgenic *Cyp1a1Cre* and *Lgr6* knock-in strains fall consistent (Fig. S9) or even tail skin, where dynamics from transgenic *lvi* are consistent with those from progenitors labelled by *Krt14* that are primed to differentiate.

There is also general concern, which is discussed in detail below, over the appropriateness of the H2BGFP mouse as a model for proliferation.

The H2BGFP strain is the best currently available tool to estimate the average rate of division of a cell population in vivo. It has been used by many groups, not only in the epidermis, but also multiple other lineages, and gives results that are consistent with a range of other assays such as EdU based lineage tracing and live imaging⁴⁻⁹. The results of H2BGFP analysis are also consistent with both pulse labelling and short-term lineage tracing with modified nucleotides, as we discuss further below^{4,8}.

Specific concerns:

Figure 2

1. More explanation is required for the H2BGFP mouse model as it relates to IFE basal layer proliferation. First, the proliferative index of mouse basal IFE across a variety of genetic backgrounds is low, ranging from 2-5%. As such a maximum of 30% of basal cells would be predicted to undergo a round of proliferation by 7 days. However, Figure 2B indicates that the entire basal layer has undergone at least one proliferation event (all cells show lower GFP intensity by confocal and FACS).

The reviewer raises the question of consistency between the H2BGFP assay and pulse labelling of S phase cells. However, we are unable to follow the argument that a labelling

index (LI) of 2-5% (typical values for mid-morning pulse labelling), means that only 30% of cells will undergo a round of cell division in 7 days.

The proportion of cycling cells that are labelled following a pulse of 3H thymidine, BrdU or EdU is equal to (S phase duration/cell cycle time), corrected for the proportion of basal cells which will differentiate without further division¹⁰. This calculation assumes that cells are not synchronised and hence are evenly distributed over cell cycle phases. However, this assumption is invalid, as LI varies dramatically over night-day cycle. For example in a study of epidermis it was measured at 13% at 5am and 5% at 10 am in back skin¹¹. Similar variation is seen in the mitotic index of the esophagus¹². The measurement of the duration of S phase is also problematic, with substantial discrepancies between the different methods used in the cell kinetic literature, see Chapter 7 of¹⁰ for a comprehensive discussion. For example, % labelled mitoses (PLM) curves may be confounded by the circadian variation in mitotic index, leading to a two-fold difference in the estimated duration of S phase depending on whether the assay is started in the morning or the evening¹².

With these significant caveats, is it feasible for our estimates of mean cycle time and the proportion of proliferating basal cells to be reconciled with a 2-5% LI in basal IFE as the reviewer highlights? For this 'sense check' we drew on mid-range estimates of S phase duration from the literature and calculated a predicted LI based on the values in Table 1. This gives us the following:

	Cycle time (h) ^a	Length S Phase (h)	% cells in S phase	% cycling basal cells ^a	Predicted LI % ^c	Observed LI (SD) %
Esophagus	58	8 ¹³	15	65	9.8	7.9 (1.4) ^b
Plantar	84 ^d	17 ¹⁴	21	53	10.9	12 (2) ⁴
Ear	112	9 ¹¹	16	54	4	4.7 (0.5) ³
Back	140	9 ¹⁵	6	61	4	5 (2) ¹⁶

a: values from H2B-GFP and lineage tracing analysis in this study

b: LI at 10 am, 1 hour post EdU i.p. injection, in *Lrig1-cre/confetti* mice in this study

c: LI calculated from (S phase duration/Cycle time) x % cycling basal cells

d: We note that our estimate of mean cycle time in the plantar epidermis is very similar to that of Lim et al (2.2/week= 76 hours) obtained by EdU analyses⁴.

While these values are broadly consistent with our analysis, our view is that the uncertainties in the estimates of S phase duration and the substantial variation in LI over the day/night cycle are so large as to preclude the inclusion of this table in the manuscript.

We conclude that the measurement of LI alone is insufficient to estimate cell turnover rates and can neither support or refute the parameter estimates in our work.

2. The additional shifts in the FACS histograms at 12 and 18 days suggest more proliferation events, totalling at least three over the course of an 18-day chase. The concept of the entire basal IFE undergoing three rounds or more of proliferation over the course of 2.5 weeks seems unlikely and does not fit with established proliferative index

rates. This needs to be addressed as it suggests that the transgenic model does not elaborate normal basal cell proliferation under homeostasis.

We apologize for the confusion caused by the presentation of the data regarding this point. The histograms under discussion were obtained by confocal imaging rather than FACS. As discussed above, the short duration of S phase in comparison with the cell cycle time and the proportion of basal cells not in cycle account for the perceived inconsistency between LI and progenitor division rates. We have clarified the protocol used to generate the histograms in the Main text, Suppl. text and in a new figure panel (**Fig. 2B**), as shown below:

B

(B) Confocal images of top-down views of basal-layer plane were acquired from epithelial wholemounts, and H2BGFP fluorescence was quantified in non-mitotic basal cell nuclei areas following image segmentation based on DAPI staining.

3. It is not clear whether the GFP+ population is gated prior to generation of the histogram plots shown in Fig 2B. A basal keratinocyte marker should be employed to demonstrate the percentage of basal keratinocytes that are GFP pos v. neg at each chase time point. As presented, the analysis in Fig. 2B appears to only show GFP+ cells and does not take into account whether any basal keratinocytes may be GFP negative (therefore undergone more rounds of proliferation than their GFP+ neighbors in the IFE).

The histograms were generated from confocal imaging on instruments with high sensitivity detectors. Basal cells were identified from their position in Z stacks spanning the entire thickness of the epidermis. The GFP fluorescence was directly measured rather than being amplified by antibody staining. We were not able to detect negative cells (DAPI positive nuclei which were GFP negative) as suggested by the reviewer, as we confined our experiments to time points where all cells were above the lower limit of detection of the instrument.

We hope the new panel added to **Figure 2**, shown above, will clarify the methodology used. Additionally, we have now included **Suppl. Videos 1-4** showing Z-stack views of the epidermis and esophagus after induction of H2BGFP, illustrating the method we use.

Fig S6

1. The GFP intensity in the Lrig1-Cre mouse model shown in Fig S6 does appear heterogenous. As such, while endogenous Lrig1 appears uniform by tissue immunofluorescence, Cre expression may differ dramatically in basal esophageal cells. This possibility confounds the Author's interpretations over an SP model if they are preferentially targeting high-expressing Cre cells with low dose tamoxifen. FACS analysis may remedy this issue.

As discussed above, the levels of *Lrig1* transcription, reported by GFP expression, are indeed somewhat heterogeneous, as is the case with most markers. However, this is not our argument for the keratinocytes being functionally equivalent progenitors. It is the tracking the evolving clone size distributions of the labelled population over months of cell divisions that drives us to conclude that the *Lrig1* labelled population represents a single population with the characteristics of SP model. The % of labelled basal cells remained approximately constant during the duration of the experiment, indicative of representative labelling of the proliferative population in a homeostatic tissue. Indeed, the common features shared between lineage tracing data sets labelled with diverse *Cre* strains analysed in our study is consistent with the predictions of the SP model.

The DAPI staining channel has now been included along with the image showing *Lrig1-GFP* fluorescence, in response to another point (see comment from reviewer #2) (**Fig. S6B**).

Reviewer #2 (Remarks to the author):

In this study Piedrafita et al. perform statistical meta-analysis of previously published as well as newly generated experimental data to test alternative hypotheses for the cellular mechanism of homeostatic maintenance of squamous stratified epithelia. The identity of stem cell and progenitor populations and the mode by which they are replenishing stratified epithelia such as the skin epidermis and the esophagus have been subjects of intense research for the past decades. Alternative interpretations of experimental data from these studies have led to mutually exclusive hypotheses, which despite the numerous high-impact publications remain unsettled and are subject of intense debate to this date. The current study by Piedrafita et al. does not offer any new ground-breaking discoveries. However, using an integrative and rigorous quantitative approach the authors attempt to explain and reconcile the apparent disparity of proposed models. For this, they take advantage of published clonal analysis data and combine them with their own lineage tracing analyses. Furthermore, the authors perform new, elegant H2B-GFP based, label-retention experiments that provide critical constrains to further refine their mathematical model. They conclude that the vast majority of experimental data in past and present studies are consistent with a “single progenitor” model which supports the idea of an equipotent population of stem cells at the basal layer of squamous stratified epithelia that support the tissue homeostasis through stochastic but balanced fates at the population level. Despite the apparent lack of novelty, this manuscript addresses an important question in the stem cell and epithelial biology fields and provides critically needed insight and clarity that in my view (and until new evidence come to light) settles the ongoing debate. For these reasons I recommend that this manuscript is published in its current form and in a timely fashion.

We are most grateful to the reviewer for their comments which capture the aims of the manuscript.

- There are other squamous stratified epithelia that do not appear to adhere to the same SP model as the skin epidermis and the esophagus. One example is the ocular surface epithelium, which is also stratified but which follows an apparent hierarchical organization with long lived stem cells located at the periphery, in the limbus while transient amplified progenitors migrate into the center of the cornea. Additionally, transitional zones in the anorectal tract appear to also have unique cellular organization and activities. Other squamous stratified epithelia, like those in the reproductive tract are less studied and therefore, poorly understood. Based on these examples I feel it is prudent to avoid blanket statements about the models tested in this study and their application to all squamous epithelia.

This comment is quite correct regarding the corneal system. We have therefore modified the Title and statements in the Abstract and Discussion to reflect that our findings are restricted to the epidermis and esophageal epithelium.

Revised Title:

“A single-progenitor model as the unifying paradigm of epidermal and esophageal epithelial maintenance.”

- Fig 1. According to table 1 the estimated rate of division is 1.2/week for the back epidermis. However, at their 7 day timepoint the fluorescence intensity seems particularly low and certainly less than 50% of that in day 0? The same seems to be the case for the examples shown in Fig.S3B. Does fixation affect the intensity and subsequent quantifications?

We are most grateful to the reviewer for pointing this out. Indeed, we have found an error and images from time 12 and 18 days batches were mistakenly labelled 7 and 12 days in the case of ear epidermis (Fig. S2D). This has now been corrected:

We have also checked that all confocal images shown for the different territories are representative of H2BGFP intensity levels for each time point. To do so, we calculated the average individual-nuclei H2BGFP intensity in every field of view and ensure all images shown fall within mean \pm SD of those from the same time point. Another image was replaced consequently to fulfil this criterion, specifically the one displayed as representative from 12 days in plantar hindpaw (Fig. S2B).

Summary statistics from the above analysis are now accessible in a new sheet in Table S2. Also, to facilitate the readout of the H2BGFP dilution process and orient on the level of field heterogeneity, we include next to histograms of individual cell H2BGFP intensities for each time point the values of the mean \pm s.e.m. across the different fields of view in Fig. 2C and Fig. S2B,D,F.

In the case of scale and interscale tail epidermis, there was a high inter-animal variation in the average H2BGFP levels, as shown in Fig. S3C, with much smaller variation in intensity levels between cells in the same field of view. For this reason, we have decided to include representative images from *each* individual mouse separately instead, i.e. 3 images per time point and territory, to highlight the variability between animals.

All samples in our study were fixed with 4% PFA prior to confocal imaging. Fixation does not detectably impact H2BGFP levels in this method.

Also, how do the authors distinguish basal cells? A month of Doxy induction means that both basal and suprabasal layers are GFP+.

Basal cells were identified from their basal location in 3 dimensional confocal z stack images^{2, 17, 18}. We have clarified this point both in Supplementary and the Main text,

‘Dox was then withdrawn and H2BGFP protein levels in individual basal keratinocytes tracked by direct, *in situ* measurement of GFP fluorescence from confocal images of epithelial wholemounts at multiple time points. We examined esophagus and epidermis from plantar area of the hind paw, ear, and tail (**Fig. 2C**; **Fig. S2**; **Fig. S3**; **Suppl. Video 1-3**). Optical sections through the deepest, basal cell layer were taken over at least 5 fields of view per tissue / animal and H2BGFP fluorescence quantified for all non-mitotic nuclei following image segmentation based on DAPI staining (**Fig. 2B**; **Suppl. Methods**).’

We have added a new panel to **Fig. 2** (new **Fig. 2B**) to show the method we used; see response to Reviewer 1.

Perhaps the authors could consider including representative serial optical sections in the form of movies accompanying the manuscript.

We now include movies showing serial optical sections (confocal Z-stacks) of the esophagus and epidermis at multiple body sites as requested.

- The label retaining experiments in Fig2/S2 are very informative and provide critical cell proliferation parameters for the subsequent mathematical modeling. It is particularly important that the authors make a specific mention and control for the presence of GFP+/CD45+, non-epithelial cells. This apparent artifact of the H2B-GFP mouse model is often overlooked in published studies but could have significant implications for the interpretation of the data.

We agree with the reviewer that excluding CD45+ intra-epidermal Langerhans cells and lymphocytes from the analysis of keratinocyte proliferation is crucial. All H2BGFP samples were stained for CD45 and positive cells excluded. We have clarified this in the revised Methods section and include a new panel (**B**) in **Fig. 2** to illustrate the protocol for H2BGFP analysis:

(B) Confocal images of top-down views of basal-layer plane were acquired from epithelial wholemounts, and H2BGFP fluorescence was quantified in non-mitotic basal cell nuclei areas following image segmentation based on DAPI staining.

In addition, we highlight some of the observed CD45+ LRCs in the confocal images shown for each territory with arrowheads (**Fig. S2B,D,F; Fig. S3B**).

- Since the computational approach carries the entire novelty of the paper it should be explained more clearly. I found the figures (both supplementary, SF4 and SF5) corresponding to the section of the paper in which this is described (“A common analytical approach integrating ...”) somewhat confusing. This may be primarily due to my lack of expertise in mathematical modelling but it was often unclear how the conclusions made in the text, for example that distribution of cell cycle lengths affects predictions of clonal growth in the short but not in the long term were supported by these figures. Perhaps the authors could refer to specific panels rather than the entire supplementary figure. Along the same lines, it’s unclear to me why these important figures detailing the main contribution of the paper are in the supplementary section.

We appreciate reviewer's comment and have included two pedagogic schemes as introductory panels to **Fig. S4** and **Fig. S5** (**Fig. S4A** and **Fig. S5A**, respectively) to address the technical points in these supplementary figures. In addition we include an overview of the analytic approach in **Figure 3** to summarise the key elements in these figures.

In **Fig. S4A** we now show how model parameter inference can be challenging without measuring the average progenitor-cell division rate, since without knowing this one could find multiple sets of parameter values that fit the same experimental data equally well. This observation is in line with the outcome from our more formal analysis shown in **Fig. S4B**.

Fig. S4 Challenges in single progenitor model parameter inference. (A-B) Uncertainty over the progenitor-cell division rate λ can affect the accuracy of inferring the value of the other model parameters (r, ρ). (A) Scheme illustrating the possible fate of different simulated clones (#1, #2, #m) under various parameter scenarios. For low ratios of r/ρ most divisions are predicted to be asymmetric, but a high value of λ results in a broadens the clone size distributions at a given time point t_{end} (grey bars). Similar effects could be expected if λ remained unchanged but r/ρ ratio increased, since this results in a higher probability of symmetric division events. Thus, different parameter values could provide adequate fits to an experimental clone size distribution (depicted as orange lines).

In **Fig. S5A** we highlight how assumptions on cell-cycle time distribution can skew parameter inferences derived from clone sizes. Again, this observation is supported by the deeper analysis shown in panels (**Fig. S5B-E**).

Fig. S5 The cell-cycle time distribution impacts parameter inference from clone size distributions. (A) Schematic illustration of clonal dynamics inferred for the SP model under

two different cell cycle time (t_{cc}) distributions of the individual cell-cycle time: exponential (*EXP*) and a delayed gamma distribution (*GAM*), where cells can only divide after a minimum refractory period τ_R (with shape controlled by parameter κ). Despite the distributions sharing the same average cell-cycle time (blue vertical line), the broad *EXP* distribution predicts that individual clones (#1, #2, #3, #m) will have a wider distribution of sizes (grey bars) after a short time chase (t_1) than the *GAM* distribution. This difference is less apparent after multiple rounds of division (t_2). Thus, the form of the cycle time distribution can impact parameter inference experimental clone sizes (orange lines) at early time points after cell labelling.

As suggested, we now refer to specific panels rather than the entire supplementary figure in the Text, and include a clarification of each observation in the corresponding paragraph. This now reads:

“By incorporating the measurement of the *average* division rate, we could reduce the uncertainty in the parameter estimation, a problem that has been generally overlooked in these stochastic models (**Fig. S4A-B**). For example, a relatively high division rate and modest proportion of symmetric division outcomes predict a similar clone size distribution to those with a slower turnover rate but higher level of symmetric divisions. In turn, whilst long-term model predictions on clone-size distributions remained largely unaffected by the assumptions on the cell-cycle time *distribution*, introducing realistic estimates for the distribution of individual cell-cycle lengths affected short-term clone-size predictions, impacting on the inferred parameter values (**Fig. S5**). This is due to the probability of a chain of consecutive division events deviating from the average rate, for example a run of several consecutive divisions shorter than average cell cycle times (**Fig. S5A**). This results in a broadening of the clone size distribution at early time points after labelling. At later times, where many rounds of division have occurred in each clone, these random cycle time variations regress towards the mean cycle time of the population (**Fig. S5A**). This disputes most analyses that use a Markovian implementation which makes the biologically implausible assumption that cell cycle times are distributed exponentially (i.e. the likeliest time for a cell to divide is immediately after the division that generated it). We therefore developed a robust quantitative approach where cell-cycle attributes estimated from H2B-GFP experiments were embodied in (non-Markovian) model simulations, and a subsequent maximum likelihood inference (MLE) method was applied across the available data sets for each body site to challenge whether each of them was consistent with the SP paradigm (**Fig. 3; Suppl. Methods**).”

Since the meaning of these two figures is thus to highlight the main technical limitations of previous analyses and they are necessarily quite detailed, we feel more comfortable keeping them as Supplementary, so as not to impair a biological reader’s ability to follow the narrative. However, to clarify our approach, we now summarise our approach in a new figure, **Fig. 3**, to clarify the process of the paper

Figure 3

Fig. 3 Method for single-progenitor model testing and parameter inference. Method to single-progenitor model testing and infer model parameters. Orange boxes indicate experiments and resulting data, grey box computational model and parameter estimation. Italics indicate parameters in the SP-model. The multimodality testing of H2B-GFP data showed that there is a single population dividing at the same average rate in epidermis and esophagus, consistent with the SP model (**Fig.2 D**). To test the SP model, the average cell cycle time (λ) and cell-cycle time distribution were inferred from H2B-GFP experiments. These values are used in computational analysis to estimate the values of the other parameters in the SP model, the proportion of progenitor cells in the basal layer ρ , the proportion of symmetric cell division outcomes r , and the stratification rate of differentiating cells leaving the basal cell layer (Γ). Multiple sets of values for the unknown parameters were tested. For each set of unknown parameter values 100,000 progenitor-derived clones were simulated (lines) and inferred clone size distributions compared to experimental ones (points) obtained from lineage tracing. The likeliest sets of parameter values were obtained by maximum likelihood estimation for each lineage tracing data set. The quality of the fit was assessed by determining whether the simulated values lie within the 95% confidence interval of the experimental clone size measurements at each time point.

- Fig.S6B It's hard to tell whether that is actually a 94% positive population. Can the authors show a picture that also has DAPI staining to have a better visualization of the ubiquitous labeling of the *Lrig-CreER* line?

We have included a picture that includes DAPI staining along with the previous one showing *Lrig1*-GFP (Fig. S6B):

We highlight *Lrig1*-GFP negative cells that correspond to bright DAPI-stained nuclei, which are indicative of mitosis (asterisks). It is likely that GFP protein levels fall during progression through mitosis.

- Fig.S6D How long after induction was this image acquired?

This image was acquired 30 days post induction. We have added this to the figure legend (Fig. S6D) as well as the scale bar that was missing.

- When the authors reanalyzed the data from Doupe et al. (2010) they claim the original authors reached a different conclusion regarding the progenitor model the dataset supported. Can the authors explain why different conclusions were reached from the same dataset in this particular case? (similar to what is discussed later in the text for Sada et al. (2016)).

We have clarified this point in the Suppl. Material. The text reads:

“In Doupe et al we first studied the validity of the SP model in the mouse esophageal epithelium, by lineage tracing using Ah-CreER R26EYFP mice. An analytical approximation was then followed to solve the theoretical clone size likelihoods and Bayesian inference used for SP model parameter estimation. The estimate for $\lambda = 2/\text{week}$ at that time was congruent with an independent H2BGFP dilution experiment, even though we now consider that the poor late-time H2BGFP signal-to-noise ratio on that occasion could lead us to underestimate the true division rate. In the present study we got a higher average division rate $\lambda = 3/\text{week}$ with new technology and more animals and time points (Fig. 2). This fact as well as the consideration of Gamma-distributed cell-cycle periods made our new parameter estimates to slightly deviate from those reported in Doupe et al (Fig. 4D,E; section 4).”

We have similarly added a comment in the Main text:

“A close fit to the SP model was obtained for very similar parameter values to the ones above, with cell-cycle time constrains resulting in an improved match over short-term clone sizes compared with fits in the original publication, **where cell-cycle time distributions were assumed exponential for simplicity (Fig. 4D-E; Fig. S7E; Table 1; Table S4; Suppl. Methods).**”

- In page3 the authors state “In contrast, in the mouse paw epidermis hair is absent but sweat ducts are common”. This statement is incorrect as hair follicles are present in varying densities in the mouse footpads (see Kamberov & Tabin, PNAS 2015).

We apologise for the lack of clarity on this point. The hind paw contains two regions, the anterior acrosyngia with the foot pads that are analysed in the work of Kamberov et al, and the posterior plantar epidermis, studied by Lim et al., which is devoid of appendages. The area we report on in this study was plantar epidermis, as defined in Lim et al.:

FIG S1A from ⁴:

We now include a cartoon to explain this point (Fig. S2B) and add a clarification in the figure legend:

Analysis of hindpaw epidermis was confined to the posterior, plantar region (pl.), excluding the acrosyngia (ac.), see cartoon.

Reviewer #3 (Remarks to the Author):

Piedrafita et al. provided a thoughtful analysis, and devised a revised model to explain how squamous epithelial tissues are maintained under homeostatic conditions. The author s described the cell-cycle properties of basal cells extracted by histone dilution data using an integrative stochastic model, and concluded that a single progenitor model can explain squamous epithelial tissue maintenance. Importantly, the authors demonstrate that their single progenitor model can explain previously proposed, mutually irreconcilable models of epidermal homeostasis. The theory behind the proposed model is consistent, and carefully explained. The manuscript is technically sound, and the quality of data presentation is high. These results will expand our understanding on how epithelial tissues are maintained under homeostatic conditions, and they represent a useful resource for further mechanistic studies and possible extrapolation to other tissues characterized by fast turn-over rates.

We are most grateful to the reviewer for their positive assessment of the work and its impact.

Here are some minor comments:

Specific comments:

1. *Can the newly revised single progenitor model be used to explain cellular competition during development and/or tissue repair? If not, what are the major new elements needed in the model? Some discussions along this line would be helpful.*

This is an important point. We and others have highlighted the need for progenitor cells local to the site of injury to transiently generate an excess of progenitor over differentiating daughters to repair the epithelial defect, as part of a coordinated process integrated with migration and altered differentiation^{2, 19, 20}. We now include the following text in the discussion to address this issue:

‘The single progenitor model captures the average behaviour of progenitor cells during homeostasis. However, epithelia are frequently subject to wounding. To repair the tissue requires a temporary imbalance in cell fate, with the progenitors close to the wound producing an excess of progenitor over differentiating daughters on average. This occurs as part of a coordinated set of responses that includes cell migration and altered cell differentiation^{2, 19, 20}. Once the epithelial defect is resolved, the progenitors revert to homeostatic balance. In esophageal epithelium and the plantar epidermis, wound repair is achieved by progenitors alone^{2, 4}. In the epidermis at other sites, cells migrating from other proliferative compartments, the hair follicles and sweat ducts, may also contribute to wound healing²¹⁻²³. The ability to transiently increase the likelihood of progenitors generating proliferating progeny provides a rapid and robust response to injury. The down side of this adjustable progenitor fate is that it may be subverted by mutations acquired during tissue aging, leading to mutant clonal expansions that may undergo malignant transformation^{18, 24-26}.’

2. *Ghazizadeh et al. (PMID: 15675956) developed a model of how human skin epidermis is*

maintained using long-term lineage tracing data in xenografts. While human epidermal maintenance is not the main aspect of this paper, is the proposed model applicable to human epidermal maintenance?

We agree with the reviewer that the applicability of our findings to humans is worth discussing. The technically challenging work of Ghazizadeh is the only study of single cell resolution lineage tracing in human epidermis that is comparable with mouse studies. We now discuss this work as follows:

‘How might these findings in mice relate to homeostasis human epidermis? Human skin differs from that of mice with many more epidermal cell layers and undulates in thickness at most body sites creating folds called rete ridges and dermal papillae²⁷. Nevertheless, a population of cells with balanced stochastic cell fate generating equal proportions of proliferating and differentiating cells has been identified in a live imaging study of human keratinocytes in primary culture¹⁹. In vivo lineage tracing in humans is not feasible. However, human epidermis has been grafted onto immune compromised mice and injected with lentiviral vectors carrying fluorescent protein reporters. When the resulting clones were imaged 6 months later they were found to vary widely in size and shape and arise from any point in the basal layer, both in rete ridges and dermal papillae²⁸. These findings are consistent with the single progenitor paradigm, but cannot provide quantitative challenge to the model available in mice.’

3. Using live imaging approach in living mice, Rompolas et al. (PMID: 27229141) suggests that a single proliferating population maintains ear and paw epidermis. This model resonates with the model proposed herein. Does those data fit with their newly revised single progenitor model?

The approach developed by Rompolas et al. was to repeatedly imaged the same clones in paw and ear skin of in a live mouse at 48 hour intervals up to 12 days⁹.

Their results are wholly consistent with our conclusions here, in particular they show that

- cell fate is stochastic
- the probability of generating progenitor and differentiated daughters is equal
- cell fate is not coordinated across generations

We have added a discussion of Rompolas et al as follows:

‘The lineage tracing approaches considered above have been enriched by live imaging studies of mouse epidermis^{9, 29}. Whilst lineage tracing reveals the average behavior of a population of proliferating cells over many cell generations, live imaging allows the fate of individual cells to be resolved. Insights gained from live imaging include showing that cell fate is stochastic, the probability of generating progenitor and differentiated daughters is equal and that the fate of cells is not co-ordinated across cell generations, all of which are key features of the SP model⁹.’

4. The term “good fit” was used multiple times in the manuscript. Is there any quantitative measurement for defining “good fit”?

We have taken a simple measure of a good fit as being that the agreement between the model with the most likely set of parameter values passes through the 95% confidence interval ($\pm 1.96 \times \text{s.e.m}$) in the data set. For example, in Figure 4E, the simulation lies within the 95% CI with all 49 data points measured in *AhcreERT R26 EYFP* clone size distribution (Lower panel), and with 27/28 data points for the *Lrig1-cre R26 confetti* strain (the one outlying point, the proportion of 5-8 cell clones at 12 weeks is also not far from the model prediction). This degree of agreement with experimental values from hundreds of clones in multiple animals is strong validation of the model.

There is no single metric to quantify fit quality that formally that takes into account both the 'plausible' range of variance in the simulation with the range of parameter values that give a statistically equivalent fit and the experimental 'error' that follows from sampling a finite number of clones and inter mouse variation. We can however quantify the degree of agreement between the optimal model prediction and the average of the experimental data.

that may be used to express the degree of agreement. We have calculated two measures of the quality of the fit, the determination coefficient, R^2 , value and the standard the standard error of the fit S . R^2 is simple and widely used but is more adapted for linear regression analysis, so we also include S which is more suited to a non-linear fit. We have calculated these values at each time point for every model fit in the paper (see supplementary table 4). We give the all-time point averages R_T^2 and S_T in the main text as convenient summary statistics.

We have added the following to the text commenting on Figure 4.

Lrig1/confetti mice, page 11:

'The model predictions were within the 95% confidence interval of the measured proportion of clones of a given size at each time point in 27/28 cases. To quantify the quality of the fit, we calculated both the correlation coefficient between the model prediction and measured clone sizes, averaged across all time points, R_T^2 , and the standard error of the fit, S_T , a measure of the standard deviation between the model estimates and the experimental data, averaged over all time points. For the fit of the SP model to the *Lrig1/Confetti* data set, $R_T^2 = 0.93$, $S_T = 4.3$. Values of R^2 and S for experimental data at each time point are given in **Supplementary Table S4**.'

Re the *AhcreR26EYFP* experiment, page 12:

'Parameter values very similar to those from the *Lrig1/confetti* experiment gave predictions from the SP model within the 95% CI for all 49 points in the experimental data set (**Fig. 4D-E; Fig. S7E; Table 1; Supplementary Table S4; Suppl. Methods**). Quantifying the quality of fit, we found $R_T^2 = 0.98$, $S_T = 2.8$. We noted that including the cell-cycle time constraints resulted in an improved agreement with early time point clone sizes compared with the original publication ($R_T^2 = 0.97$, $S_T = 3.3$), where cell-cycle time distributions were assumed exponential (see **Supplementary Table S4** for detailed goodness-of-fit statistics) ⁵.'

We also include detailed R^2 and S values for all comparisons in the text in Supplementary table S4, citing the summary R_T^2 and S_T values in the main text.

5. In Figure 3B, no data for two clones derived from *Lrig1/Confetti* mice is shown for 180 days. Is there a particular reason for why these were not shown?

Unfortunately, we do not have CFP or YFP 180 day images (this was due to a staining step at this time point that destroyed the Cyan and Yellow fluorescent protein signal). We do not have an experiment running at present so are unable to generate replacement images in a timely manner. We would be happy to add a note to the figure legend saying: 'CFP and YFP clones could not be imaged at 180 days due to technical problems.'. However, we opted to remove in this figure panels showing CFP and YFP clones from earlier time points as well, to keep just images of RFP clones along all time points for clarity. Then, we show typical RFP, CFP and YFP clones from a given time point (84 days) in new panel **Fig. S7A**.

6. In Figure 3C, it would be useful to present: i) average clone size, ii) number of clones/mm², and iii) percentage of basal cells per individual clones (i.e. CFP, YFP, and RFP).

We have added the data requested. These are shown in **Fig. S7B**, as they fit well with the confocal images of individual RFP, CFP, YFP clones at 84 days presented in the same figure (**Fig. S7A**).

Trends by individual reporters were similar as those shown by a merged analysis, with average clone size increasing, the density of surviving clones decreasing, and the %labelled basal cells remaining approximately constant, even though sample sizes got limited for individual fluorescent reporters at latest time points (see comment on legend below). We now cite these panels in the Suppl. Methods, along with their statistical analysis: "No statistical differences in the distributions of clone size frequencies were seen between CFP, RFP and YFP labelled cell populations at any given time (Kruskal-Wallis test, $p = 0.17, 0.27, 0.99, 0.22$ at time 10d, 30d, 84d, 180d, respectively; non-significant too by pairwise comparisons using Kolmogorov-Smirnov test)". In this way we show the results from a merged analysis in the Main figure (**Fig. 4C**). We would be happy, though, to move the individual panels of each reporter to Main figure if the reviewer considered it more convenient.

(B) Quantitative traits of RFP-, CFP-, and YFP- labelled clone populations over time: average basal-layer clone size (i.e. mean number of basal cells/surviving clone) (top panels), average density of labelled clones in the basal layer (middle panels), average fraction of labelled basal cells (bottom panels) at the indicated time points. Observed values (dots) and error bars (mean \pm s.e.m.) from $n \geq 3$ animals. Straight colored lines and shading in bottom panels indicate mean and s.e.m. across all time points, consistent with homeostatic behavior (only CFP clones were undetected in the cohort of mice used for clone density and % labelled basal cell quantifications at latest time point, probably due to low initial induction and limited sampling).

We have similarly added a comment in the Main text that reads:

“The pooled *Confetti* clone data set displayed several important features, which were recapitulated by clones labelled with each individual reporter. No statistically significant differences were observed between CFP, YFP and RFP clone size distributions at each time point: see **Suppl. Methods**. The density (clones/area) of labelled clones decreased progressively, consistent with clone loss through differentiation, while the number of basal and suprabasal cells in the remaining clones rose (**Fig. 4C**; **Fig. S7B-C**).”

7. The H2BGFP dilution analysis and keratinocyte cell cycle inference in dorsal skin differs from that of paw, ear and tail (related to Supplementary Figure S2 and S3). Is there a reason why dilution experiments were performed differently in back skin compared to other skin territories?

The dorsal skin experiments were done in a similar way but were controls for another project we published last year¹⁸. This is now indicated in the **Suppl. Methods**. The time points were thus taken to fit in with a different set of experiments, but overall, the time window overlaps with that in the present study, so we feel it appropriate to include this data. Both studies were done in an identical mouse strain on the same genetic background, under conditions of homeostasis, and following same methodology.

8. *Supplementary Figure S4D shows that the single progenitor model parameter discrimination improves as the sample size is increased. Here, the legend suggests that 100, 1000, and 10000 clones were sampled but figure suggests 400, 4000, and 40000. What was the sample size used?*

We used 100, 1000 and 10000 clones per time point, and 4 time points in total. We have clarified this in the revised legend that reads:

“(D) SP model parameter discrimination improves as increasing the sample size. The same synthetic data sets described above were used, and inference analyses repeated by randomly sampling 100, 1,000 and 10,000 clones per time point at four time points (1, 2, 4, and 6 weeks), i.e. a total of $N = 400, 4,000$ and $40,000$ clones. Results are shown as bootstrapping averages.”

9. *In supplemental material, the “P” in Eq. (15) and Eq. (16), are they the same? The descriptions for Non-Markovian simulations are confusing.*

They refer to different probabilities: The first is the probability that the cell cycle lasts a certain time (full period length). The second specifies the probability of seeing a cell that has remained a certain time without dividing, i.e. it represents a fraction of the full period and thus this probability always has to be smaller or equal to first one. We now use two different notations: P in Eq. (15) and ϕ in Eq.(16).

10. *What is “scaling behavior”?*

In our context, we observe scaling of the basal layer clone size distributions (not only the average clone size but also their degree of dispersion and shape) with time. Simply put, this means that if time doubles, not only the average but the entire distribution of clones, its shape and breadth also doubles. Klein and Simons³⁰ describe scaling as follows “The clone size distributions are predicted to have a special property, known as ‘scaling’, whereby their shape is preserved over time post-labelling, despite the average clone size growing steadily. This effectively means that the clone size distributions should ‘stretch out’ over time...”.

Mathematically, the probability of finding clones larger than a certain multiple m of the average, becomes a simple exponential with m (time invariant). This exponential is called the ‘scaling function’ and the formula is given as:

$$P_n^{cum}(t) = \exp[-n/\langle n \rangle_{surv}(t)]$$

We now reference Klein and Simons in the Main text and Supplementary Methods and have revised the main text as follows

“At late time points, the clone size distribution scaled with time. This means that if, for example, time doubles, not only the average clone size shape and breadth of the clone size distribution also double. More formally, the probability of seeing clones larger than x times the average clone size became time-invariant, following a simple exponential $f(x) = e^{-x}$ (Fig. S7D)³⁰.”

11. *Page 10 of main text: $f(x) = e^{-x}$ not $f(x)=e-x$.*

Thank you, this has now been corrected.

12. Page 12 of main text: Letter "F" in the middle of the page.

Thank you, this has now been corrected.

13. Where is the legend for subfigure E in Figure 3? It seems missing.

Thank you for pointing this out, the legend has now been added (Fig. 4E).

14. In Supplement section 2, what is cumulative frequencies? Pncum ?

Pncum is the term used for cumulative frequencies, i.e. probability of finding a clone larger than n basal cells; i.e. all clones containing $(n + 1)$, $(n + 2)$, $(n + 3)$, etc basal cells. This is now clarified in the Supplement.

Reviewer response references

1. Clayton, E. *et al.* A single type of progenitor cell maintains normal epidermis. *Nature* **446**, 185-189 (2007).
2. Doupe, D.P. *et al.* A single progenitor population switches behavior to maintain and repair esophageal epithelium. *Science* **337**, 1091-1093 (2012).
3. Doupe, D.P., Klein, A.M., Simons, B.D. & Jones, P.H. The ordered architecture of murine ear epidermis is maintained by progenitor cells with random fate. *Dev Cell* **18**, 317-323 (2010).
4. Lim, X. *et al.* Interfollicular epidermal stem cells self-renew via autocrine Wnt signaling. *Science* **342**, 1226-1230 (2013).
5. Wilson, A. *et al.* Hematopoietic Stem Cells Reversibly Switch from Dormancy to Self-Renewal during Homeostasis and Repair. *Cell* **135**, 1118-1129 (2008).
6. Watson, Julie K. *et al.* Clonal Dynamics Reveal Two Distinct Populations of Basal Cells in Slow-Turnover Airway Epithelium. *Cell reports* **12**, 90-101 (2015).
7. Foudi, A. *et al.* Analysis of histone 2B-GFP retention reveals slowly cycling hematopoietic stem cells. *Nature biotechnology* **27**, 84-90 (2009).
8. Andersen, M.S. *et al.* Tracing the cellular dynamics of sebaceous gland development in normal and perturbed states. *Nature cell biology* **21**, 924-932 (2019).
9. Rompolas, P. *et al.* Spatiotemporal coordination of stem cell commitment during epidermal homeostasis. *Science* **352**, 1471-1474 (2016).
10. Wright, N. & Alison, M.R. *The biology of epithelial cell populations*, Vol. 1. (Oxford University Press, Oxford; 1984).
11. Potten, C.S. Epidermal cell production rates. *J Invest Dermatol* **65**, 488-500 (1975).
12. Burns, E.R., Scheving, L.E., Fawcett, D.F., Gibbs, W.M. & Galatzan, R.E. Circadian influence on the frequency of labeled mitoses method in the stratified squamous epithelium of the mouse esophagus and tongue. *The Anatomical record* **184**, 265-273 (1976).
13. Thrasher, J.D. Age and the cell cycle of the mouse esophageal epithelium. *Experimental gerontology* **6**, 19-24 (1971).

14. Laurence, E.B. The epidermal chalone and keratinizing epithelia. *National Cancer Institute monograph* **38**, 61-68 (1973).
15. Clausen, O.P. Regenerative proliferation of mouse epidermal cells following application of a skin irritant (cantharidin). Flow microfluorometric DNA measurements and [3H]TdR incorporation studies of isolated basal cells. *Cell and tissue kinetics* **12**, 135-144 (1979).
16. Chapellier, B. *et al.* Physiological and retinoid-induced proliferations of epidermis basal keratinocytes are differently controlled. *EMBO J* **21**, 3402-3413 (2002).
17. Frede, J., Greulich, P., Nagy, T., Simons, B.D. & Jones, P.H. A single dividing cell population with imbalanced fate drives oesophageal tumour growth. *Nat Cell Biol* **18**, 967-978 (2016).
18. Murai, K. *et al.* Epidermal Tissue Adapts to Restrain Progenitors Carrying Clonal p53 Mutations. *Cell Stem Cell* **23**, 687-699.e688 (2018).
19. Roshan, A. *et al.* Human keratinocytes have two interconvertible modes of proliferation. *Nat Cell Biol* **18**, 145-156 (2016).
20. Park, S. *et al.* Tissue-scale coordination of cellular behaviour promotes epidermal wound repair in live mice. *Nat Cell Biol* **19**, 155-163 (2017).
21. Levy, V., Lindon, C., Zheng, Y., Harfe, B.D. & Morgan, B.A. Epidermal stem cells arise from the hair follicle after wounding. *Faseb J* **21**, 1358-1366 (2007).
22. Ito, M. *et al.* Stem cells in the hair follicle bulge contribute to wound repair but not to homeostasis of the epidermis. *Nature medicine* **11**, 1351-1354 (2005).
23. Lu, C.P. *et al.* Identification of stem cell populations in sweat glands and ducts reveals roles in homeostasis and wound repair. *Cell* **150**, 136-150 (2012).
24. Fernandez-Antoran, D. *et al.* Outcompeting p53-Mutant Cells in the Normal Esophagus by Redox Manipulation. *Cell stem cell* (2019).
25. Alcolea, M.P. *et al.* Differentiation imbalance in single oesophageal progenitor cells causes clonal immortalization and field change. *Nat Cell Biol* **16**, 615-622 (2014).
26. Alcolea, M.P. & Jones, P.H. Cell competition: Winning out by losing notch. *Cell Cycle* **14**, 9-17 (2015).
27. Lavker, R.M. & Sun, T.T. Heterogeneity in epidermal basal keratinocytes: morphological and functional correlations. *Science* **215**, 1239-1241 (1982).
28. Ghazizadeh, S. & Taichman, L.B. Organization of stem cells and their progeny in human epidermis. *J Invest Dermatol* **124**, 367-372 (2005).
29. Mesa, K.R. *et al.* Homeostatic Epidermal Stem Cell Self-Renewal Is Driven by Local Differentiation. *Cell Stem Cell* **23**, 677-686.e674 (2018).
30. Klein, A.M. & Simons, B.D. Universal patterns of stem cell fate in cycling adult tissues. *Development* **138**, 3103-3111 (2011).

Reviewers' Comments:

Reviewer #1:

Remarks to the Author:

The Authors have sufficiently addressed all concerns.

Reviewer #2:

Remarks to the Author:

For the re-submission the authors went above and beyond to thoroughly address all our questions, suggestions and criticisms. The revised manuscript has been significantly improved and I therefore recommend that it is accepted for publication in its current form and without further revisions.

Reviewer #3:

Remarks to the Author:

My comments have been well addressed in the revision.